# Pareto-Optimality, Smoothness, and Stochasticity in Learning-Augmented One-Max-Search

**Ziyad Benomar** [* 1 2]   **Lorenzo Croissant** [* 1 2]   **Vianney Perchet** [1 2 3]   **Spyros Angelopoulos** [4]

## Abstract

One-max search is a classic problem in online decision-making, in which a trader acts on a sequence of revealed prices and accepts one of them irrevocably to maximize its profit. The problem has been studied both in probabilistic and in worst-case settings, notably through competitive analysis, and more recently in learning-augmented settings in which the trader has access to a *prediction* on the sequence. However, existing approaches either lack *smoothness*, or do not achieve optimal worst-case guarantees: they do not attain the best possible trade-off between the *consistency* and the *robustness* of the algorithm. We close this gap by presenting the first algorithm that simultaneously achieves both of these important objectives. Furthermore, we show how to leverage the obtained smoothness to provide an analysis of one-max search in stochastic learning-augmented settings which capture randomness in both the observed prices and the prediction.

## 1. Introduction

Recent and rapid advances in machine learning have provided the ability to learn complex patterns in data and time series. These advancements have given rise to a new computational paradigm, in which the algorithm designer has the capacity to incorporate a *prediction* oracle in the design, the theoretical analysis, and the empirical evaluation of an algorithm. The field of *learning-augmented* algorithms was born out of this emerging requirement to leverage ML techniques towards the development of more efficient algorithms.

Learning-augmented algorithms have witnessed remarkable

growth in recent years, starting with the seminal works (Lykouris & Vassilvtiskii, 2018) and (Purohit et al., 2018), particularly in *online* decision making. In this class of problems, the input is a sequence of items, which are revealed one by one, with the algorithm making an irrevocable decision on each. Here, the prediction oracle provides some inherently imperfect information on the input items, which the algorithm must be able to leverage in a judicious manner.

One of the most challenging aspects of learning-augmented (online) algorithms is their theoretical evaluation. Unlike the prediction-free setting, in which worst-case measures such as the *competitive ratio* (Borodin & El-Yaniv, 2005) evaluate algorithms on a single metric, the analysis of learning-augmented settings is multifaceted and must incorporate the effect of the prediction *error* to be meaningful. Typical desiderata (Lykouris & Vassilvtiskii, 2018) include: an efficient performance if the prediction is accurate (*consistency*); a performance that is not much worse than the competitive ratio if the predictions are arbitrarily inaccurate (*robustness*); and between these, a smooth decay of performance as the prediction error grows (*smoothness*). This marks a significant departure from the worst-case, and overly pessimistic competitive analysis, and allows for a much more nuanced and *beyond worst-case* performance evaluation.

Achieving all these objectives simultaneously is a challenging task, and is even impossible for certain problems (Elenter et al., 2024). To illustrate such challenges with an example, consider the *one-max-search* problem, which models a simple, yet fundamental setting in financial trading. Here, the input is a sequence of *prices* $(p_i)_{i=1}^n \in [1, \theta]$, where $\theta$ is a known upper bound, and the online algorithm (i.e., the trader) must decide, irrevocably, which price to accept.

Under standard competitive analysis, which compares the algorithm's accepted price to the maximum price $p^* := \max_i p_i$ on a worst-case instance, the problem admits a simple, yet optimal, algorithm (El-Yaniv, 1998). In contrast, the learning-augmented setting, in which the algorithm has access to a prediction of $p^*$, is far more complex. Specifically, Sun et al. (2021) gave a *Pareto-optimal* algorithm, i.e. one that achieves the best possible trade-off between consistency and robustness. However, this algorithm lacks smoothness, which results in *brittleness*. Namely, Benomar

---

[*]Equal contribution  [1]CREST, ENSAE, Palaiseau, France [2]INRIA Fairplay joint team, Palaiseau, France [3]Criteo AI Lab, Paris, France [4]CNRS and International Laboratory on Learning Systems, Montreal, Canada. Correspondence to: Ziyad Benomar <ziyad.benomar@ensae.fr>.

*Proceedings of the 42nd International Conference on Machine Learning*, Vancouver, Canada. PMLR 267, 2025. Copyright 2025 by the author(s).

& Perchet (2025) showed that even if the prediction error is arbitrarily small, the algorithm's performance may degrade dramatically, and collapses to the robustness guarantee. This renders the algorithm unsuitable for any practical applications since perfect oracles do not exist in the real world. Benomar & Perchet (2025) addressed the issue of brittleness by using randomization: this results in an algorithm with smoother behaviour, albeit at the cost of deviating from the consistency-robustness Pareto front. In a similar vein, Angelopoulos et al. (2022) gave a smooth algorithm without any guarantees on the trade-off between consistency and robustness. Sun et al. (2024) studied the problem under a model of uncertainty-quantified predictions, in which the algorithm has access to additional, and more powerful, probabilistic information about the prediction error.

The following natural question arises: *is there an all-around optimal algorithm, that is simultaneously Pareto-optimal and smooth, and does not rely on randomisation or probabilistic assumptions about the quality of the prediction?*

### 1.1. Main contributions

Our main result answers the above question in the affirmative by giving a deterministic Pareto-optimal algorithm with smooth guarantees. Furthermore, we demonstrate how to leverage smoothness so as to extend this analysis to stochastic settings in which the input and prediction are random.

In previous works on learning-augmented one-max-search (Sun et al., 2021; Benomar & Perchet, 2025), the proposed algorithms select the first price that exceeds a *threshold* $\Phi(y)$, which is a function of the prediction $y$ of the maximum price $p^*$ in the sequence. We revisit the problem by first characterizing the class $\mathcal{P}_r$ of all consistency-robustness Pareto-optimal thresholds $\Phi$. Next, we focus on a specific family of Pareto-optimal thresholds within $\mathcal{P}_r$ which generalise the algorithm of Sun et al. (2021) but also exhibit smoothness guarantees. In particular, our analysis quantifies smoothness in this family, showing it to be inversely proportional to the maximal slope of the corresponding threshold. Guided by this insight, we find the threshold that maximizes smoothness within the class $\mathcal{P}_r$.

Furthermore, this quantification of smoothness allows to establish a near-matching lower bound. Specifically, we show that, for a multiplicative definition of the error, no Pareto-optimal algorithm can guarantee better smoothness than our algorithm, for a large range of robustness values. For the additive definition of the prediction error, which is commonly used, we show that our algorithm optimises smoothness for all robustness values, thus attaining the *triple* Pareto front of consistency, robustness and smoothness.

The combination of smoothness and Pareto-optimality of our family of thresholds has direct practical benefits in han-

dling the real-world uncertainty of predictions. When predictions and prices are both tied to a random environment (e.g. a financial market), we show how to derive general form bounds in expectation as a function of the distributions of predictions and prices and, for the first time, of their *coupling*. We provide prediction-quality metrics which help us better capture the notion of the "usefulness" of a prediction in stochastic environments and give detailed bounds on concrete settings. We also provide a general framework of analysis based on optimal transport. The use of optimal transport for competitive analysis in stochastic contexts is novel and opens new and interesting research perspectives.

We validate our theoretical results through numerical experiments, in which we compare our algorithm to the state of the art, by testing it under both synthetic and real data.

### 1.2. Related work

**Learning-augmented algorithms.** Algorithms with predictions have been studied in a large variety of online problems, such as rent-and-buy problems (Gollapudi & Panigrahi, 2019), scheduling (Lattanzi et al., 2020), caching (Lykouris & Vassilvtiskii, 2018), matching (Antoniadis et al., 2020), packing (Im et al., 2021), covering (Bamas et al., 2020) and secretaries Dütting et al. (2024). This paradigm also has applications beyond online computation, and has been used to improve the runtime of algorithms for classical problems such as sorting (Bai & Coester, 2023) and graph problems (Azar et al., 2022), as well as for the design of data structures such as search trees (Lin et al., 2022), dictionaries (Zeynali et al., 2024), and priority queues (Benomar & Coester). We emphasize that the above lists only some representative works, and we refer to the online repository of Lindermayr & Megow (2025).

**Pareto-optimal algorithms.** Several studies have focused on consistency-robustness trade-offs in learning-augmented algorithms, e.g. (Sun et al., 2021; Wei & Zhang, 2020; Lee et al., 2024; Angelopoulos, 2023; Bamas et al., 2020; Christianson et al., 2023; Almanza et al., 2021). However, Pareto-optimality imposes constraints which may, in certain cases, compromise smoothness. The brittleness of Pareto-optimal algorithms for problems such as one-way trading was observed by Elenter et al. (2024), who proposed a user-defined approach to smoothness, and by Benomar & Perchet (2025) who relied on randomization. These approaches differ from ours, in that the profile-based framework of Elenter et al. (2024) does not always lead to an objective and measurable notion of consistency. Moreover, we show that randomization is not necessary to achieve Pareto optimality.

**One-Max Search.** El-Yaniv (1998) showed that the optimal competitive ratio of (deterministic) one-max-search is $1/\sqrt{\theta}$, under the assumption that each price in the se-

quence is in $[1, \theta]$, where $\theta$ is a known upper bound on $p^*$. This assumption is required in order to achieve a bounded competitive ratio and has remained in all subsequent works on learning-augmented algorithms for this problem, such as (Sun et al., 2021; Angelopoulos et al., 2022; Sun et al., 2024; Benomar & Perchet, 2025). The randomized version of one-max-search is equivalent to the *one-way trading* problem, in which the trader can sell fractional amounts. The optimal competitive ratio, for this problem, is $O(1/\log \theta)$ (El-Yaniv, 1998). Pareto-optimal algorithms for one-way trading were given by Sun et al. (2021), however Elenter et al. (2024) showed that any Pareto-optimal algorithm for this problem is brittle and thus cannot guarantee smoothness. One-max-search and one-way trading model fundamental settings of trading, and many variants and generalizations have been studied under the competitive ratio, see the survey (Mohr et al., 2014). One must note that worst-case measures such as the competitive ratio aim to model settings in which no Bayesian assumptions are known to the algorithm designer. There is a very rich literature on optimal Bayesian search, see, e.g. (Rosenfield & Shapiro, 1981).

## 2. Preliminaries

In the standard setting of the one-max-search problem, the input consists of a (unknown in advance) sequence of prices $p := (p_i)_{i=1}^n \in [1, \theta]^n$, where the maximal range $\theta$ is known. At each step $i \in [n]$, the algorithm must decide irrevocably whether to accept $p_i$, terminating with a payoff of $p_i$, or to forfeit $p_i$ and proceed to step $i + 1$. If no price has been accepted by step $n$, then the payoff defaults to 1.

The *competitive ratio* of an algorithm is defined as the worst-case ratio (over all sequences $p$) between the algorithm's payoff and $p^*$, the maximum price in the sequence. A natural approach to this problem is to use threshold-based algorithms, which select the first price that exceeds a predetermined threshold $\Phi \in [1, \theta]$. We denote such an algorithm by $\mathsf{A}_\Phi$. In particular, the optimal deterministic competitive ratio is $1/\sqrt{\theta}$ and it is achieved by $\mathsf{A}_{\sqrt{\theta}}$ (El-Yaniv, 1998). Focusing on this class of algorithms is not restrictive, as in worst-case instances any deterministic algorithm performs equivalently to a threshold algorithm (see Appendix B).

In the learning-augmented setting, the decision-maker receives a *prediction* $y$ of the maximum price in the input $p$. The payoff of an algorithm $\mathsf{ALG}$ in this setting is denoted by $\mathsf{ALG}(p, y)$. In this context, threshold rules are defined as mappings $\Phi : [1, \theta] \to [1, \theta]$ that depend on the prediction. We use again $\mathsf{A}_\Phi$ to denote the corresponding algorithm. We denote by $c(\mathsf{ALG})$ and by $r(\mathsf{ALG})$ the consistency and the robustness of the algorithm, respectively, defined as

$$c(\mathsf{ALG}) = \inf_p \frac{\mathsf{ALG}(p, p^*)}{p^*} \ , \ r(\mathsf{ALG}) = \inf_{p, y} \frac{\mathsf{ALG}(p, y)}{p^*} \ .$$

Sun et al. (2021) established the Pareto front of the consistency-robustness trade-off, albeit using a different convention (the inverse ratio $p^*/\mathsf{ALG}$) for the competitive ratio[1]. Under our convention, their results show that the Pareto front of consistency and robustness is the curve:

$$\left\{ cr\theta = 1 \text{ for } (c, r) \in [\theta^{-1/2}, 1] \times [\theta^{-1}, \theta^{-1/2}] \right\} . \quad (1)$$

In contrast, we are interested not only in consistency-robustness Pareto-optimality but also in smoothness, namely in the performance as a function of the prediction *error* $\eta(p^*, y) := |p^* - y|$. An algorithm is called *smooth* if the ratio $\mathsf{ALG}(p, y)/p^*$ is lower bounded by a (non-constant) continuous and monotone function of the error $\eta(p^*, y)$.

## 3. Pareto-Optimal and Smooth Algorithms

In this section, we present our main result in regards to deterministic learning augmented algorithms, namely a Pareto-optimal and smooth family of algorithms for one-max-search. Our approach is outlined as follows. We begin by characterising the class of all thresholds $\mathcal{P}_r$ which induce Pareto-optimal algorithms (Theorem 3.1). We then present a family of thresholds in $\mathcal{P}_r$, parametrised by a value $\rho \in [0, 1]$ ( Eq. (2)) that characterises their smoothness and we show that $\rho = 1$ yields the best smoothness guarantees. We complement this result with Theorem 3.3, which shows that not only is our algorithm smooth, but any Pareto-optimal algorithm cannot improve on its smoothness.

Before we discuss our algorithms, we note that the randomized algorithm of (Benomar & Perchet, 2025) has a measurable and significant deviation from the Pareto front, even in comparison to deterministic algorithms; see Appendix D.3 for the expression of the deviation. Furthermore, the guarantees of their algorithm hold in expectation only, whereas the results we obtain do not rely on randomisation.

We now proceed with the technical statements. Theorem 3.1 below provides a characterization of all thresholds that yield Pareto-optimal levels of consistency and robustness.

**Theorem 3.1.** *For any fixed of robustness $r$, the set of all thresholds $\Phi : [1, \theta] \to [1, \theta]$ such that $\mathsf{A}_\Phi$ has robustness $r$ and consistency $1/r\theta$ is*

$$\mathcal{P}_r := \Big\{ \Phi \ : \forall z \in [1, \theta] : r\theta \le \Phi(z) \le \frac{1}{r}$$
$$\forall z \in [r\theta, \theta] : \frac{z}{r\theta} \le \Phi(z) \le z \Big\}.$$

Figure 1 illustrates the set $\mathcal{P}_r$ (shaded).

We prove the theorem via a double inclusion. First, if a threshold function $\Phi$ belongs to $\mathcal{P}_r$, then we use the bounds

---

[1]The convention that the competitive ratio of the maximization problem is in $(0, 1]$ allows for cleaner bounds on the performance as a function of the prediction error.

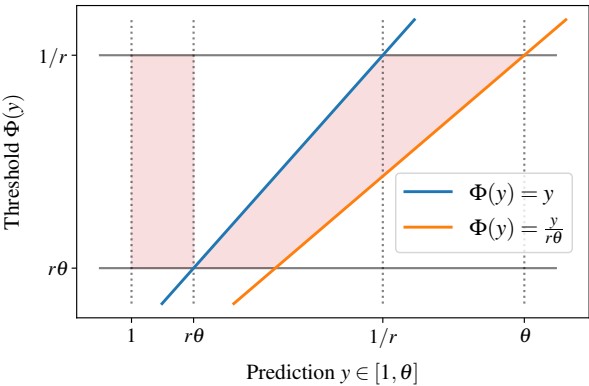

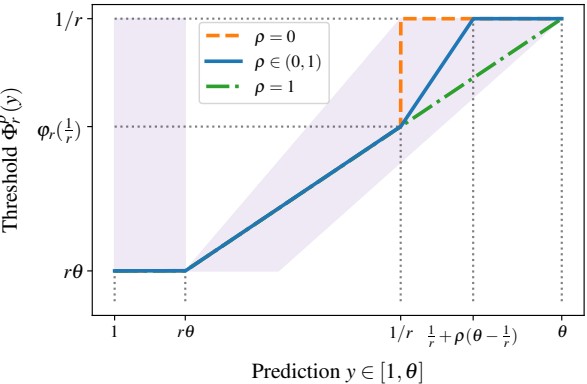

*Figure 1.* A depiction of the set $\mathcal{P}_r$ of Theorem 3.1. A threshold $\Phi$ is $r$-robust and $1/r\theta$-consistent if and only if it is a function whose graph lies in the shaded area which defines $\mathcal{P}_r$.

*Figure 2.* The threshold functions $\Phi_r^\rho$ of $\mathsf{A}_r^\rho$, as defined by Eq. (2). Note that all are equal on $[1, r^{-1})$.

on $\Phi$ defining $\mathcal{P}_r$ to prove its Pareto-optimal robustness and consistency. Conversely, for a Pareto-optimal $\Phi$, we analyse the worst-case instances $\mathcal{I}_n(q)$ (Equation (15)), where prices increase uniformly from 1 to $q$ before dropping to 1. By evaluating the ratio $\mathsf{ALG}(p, y)/p^*$ on these instances for well-chosen $q$, we show that $\Phi \in \mathcal{P}_r$.

We now turn to identifying smooth algorithms within the class $\mathcal{P}_r$. Let us begin by giving the intuition behind our approach. Our starting observation is that the algorithm of Sun et al. (2021) uses the threshold

$$\Phi_r^0(y) := \begin{cases} r\theta & \text{if } y \in [1, r\theta) \\ \varphi_r(y) & \text{if } y \in [r\theta, 1/r) \\ 1/r & \text{if } y \in [1/r, \theta] \end{cases},$$

wherein

$$\varphi_r : z \mapsto \frac{r\theta - 1}{1 - r} + \frac{1 - r^2\theta}{1 - r} \cdot \frac{z}{r\theta}$$

is the line defined by $(r\theta, r\theta)$ and $(\theta, 1/r)$. The function $\Phi_r^0$ is illustrated in Figure 2, in dashed orange. The analysis of Benomar & Perchet (2025) revealed that the brittleness of this algorithm arises from the discontinuity of $\Phi_r^0$ at the point $1/r$, as illustrated in Figure 2. This observation suggests that the smoothness of an algorithm $\mathsf{A}_\Phi$ is influenced by the maximal slope of the function $z \mapsto \Phi(z)$. To confirm this intuition, we analyse a family of algorithms $\{\mathsf{A}_r^\rho\}_{\rho \in [0,1]}$, associated with the thresholds $\{\Phi_r^\rho(y)\}_\rho$ defined by

$$\begin{cases} r\theta & \text{if } y \in [1, r\theta) \\ \varphi_r(y) & \text{if } y \in [r\theta, \frac{1}{r}) \\ \varphi_r(\frac{1}{r}) + \frac{\frac{1}{r} - \varphi_r(\frac{1}{r})}{\rho(1 - r)} \cdot \frac{y}{r\theta} & \text{if } y \in [\frac{1}{r}, \frac{1}{r} + \rho(\theta - \frac{1}{r})) \\ 1/r & \text{if } y \in [\frac{1}{r} + \rho(\theta - \frac{1}{r}), \theta] \end{cases} \quad (2)$$

Figure 2 illustrates the threshold functions $(\Phi_r^\rho)_{\rho \in [0,1]}$. Notably, the case $\rho = 0$ corresponds to the algorithm of Sun et al. (2021), while at the other extreme ($\rho = 1$) we obtain the threshold $\Phi_r^1 : z \mapsto \max(r\theta, \varphi_r(z))$.

The standard definition of smoothness involves demonstrating a continuous degradation of algorithm performance as a function of the prediction error

$$\eta(p^*, y) := |p^* - y|.$$

A key limitation of $\eta$ is its sensitivity to rescaling—multiplication of the instance by a constant factor—which makes it less suitable in the context of competitive analysis. A common solution is to express bounds in terms of the relative error

$$\frac{\eta(p^*, y)}{p^*} = \left| 1 - \frac{y}{p^*} \right|,$$

as is often done in prior work on learning-augmented algorithms. However, this introduces another complication: the asymmetry between $y$ and $p^*$.

To overcome these issues, we use a multiplicative error measure that retains the desirable properties of scale invariance and symmetry:

$$\mathcal{E} : (p^*, y) \mapsto \min\left\{ \frac{p^*}{y}, \frac{y}{p^*} \right\} \in [\theta^{-1}, 1]. \quad (3)$$

This error measure has previously been used in the context of mechanism design with predictions (Balkanski et al., 2024). Note that a perfect prediction corresponds to $\mathcal{E}(p^*, y) = 1$. We primarily focus on $\mathcal{E}$ in the remainder of the paper. However, we also provide smoothness results using the additive error measure $\eta$ for completeness, in Section 3.2.

### 3.1. Multiplicative error $\mathcal{E}$

This first theorem establishes smoothness guarantees of the family $\{A_r^\rho\}_{r,\rho}$ with respect to the error measure $\mathcal{E}$.

**Theorem 3.2.** *The family $\{A_r^\rho\}_{r,\rho}$ satisfies*

$$\frac{A_r^\rho(p,y)}{p^*} \geq \max\left(r, \frac{1}{r\theta}\mathcal{E}(p^*,y)^{s_\rho}\right) , \qquad (4)$$

*with $s_\rho := \max\left(1, \frac{1}{\rho}\left(\frac{\ln\theta}{\ln(r\theta)} - 2\right)\right)$, for $\rho \in [0,1]$.*

For $\rho \in [0,1]$, the threshold defining $A_r^\rho$ lies in $\mathcal{P}_r$, ensuring $r$-robustness and $1/r\theta$-consistency (Theorem 3.1). To establish smoothness, we determine the best value of $s$ such that $A_r^\rho/p^* \geq (1/r\theta)\mathcal{E}^s$ across all price sequences and predictions. Since the threshold is piecewise-defined, it suffices to analyse cases based on $y$ and whether the maximum price is above or below the threshold. The final smoothness guarantee is then determined by the worst such case.

The smoothness in Theorem 3.2 is quantified by the exponent $s_\rho$ of $\mathcal{E}$. A smaller value of $s_\rho$ results in slower degradation of the bound as a function of prediction error, i.e. improved smoothness. In the limit $\rho \to 0$, the exponent $s_\rho$ becomes arbitrarily large, which results in extreme sensitivity to prediction errors, i.e. brittleness. In contrast, the best smoothness is achieved by $\rho = 1$, yielding an exponent

$$s := s_1 = \max\left(1, \frac{\ln\theta}{\ln(r\theta)} - 2\right) .$$

The above positive result naturally raises the question: is the smoothness guarantee of Theorem 3.2 optimal among Pareto-optimal algorithms? This question is addressed by Theorem 3.3 which gives a lower bound on the smoothness achievable by any Pareto-optimal algorithm.

**Theorem 3.3.** *Let $A$ be any deterministic algorithm with robustness $r$ and consistency $1/r\theta$. Suppose that $A$ satisfies for all $p \in [1,\theta]^n$ and $y \in [1,\theta]$ that*

$$\frac{A(p,y)}{p^*} \geq \max\left(r, \frac{1}{r\theta}\mathcal{E}(p^*,y)^u\right) \qquad (5)$$

*for some $u \in \mathbb{R}$, then necessarily $u \geq \frac{\ln\theta}{\ln(r\theta)} - 2$.*

We first prove that, on worst-case instances $\mathcal{I}_n(q)$, any Pareto-optimal algorithm behaves like a threshold algorithm. Using results from Theorem 3.1, we establish that its threshold function must belong to $\mathcal{P}_r$. Finally, using the definition of $\mathcal{P}_r$ with well-chosen $q$ and $y$, we derive lower bounds on the smoothness $s$.

This lower bound shows the optimality of the exponent achieved by $A_r^1$ for $r \leq \theta^{-2/3}$. Indeed, if this condition is satisfied, then Theorem 3.2 gives

$$\frac{A_r^1(p,y)}{p^*} \geq \max\left(r, \frac{1}{r\theta}\mathcal{E}(p^*,y)^{\frac{\ln\theta}{\ln(r\theta)} - 2}\right) .$$

This implies that, for $r \in [\theta^{-1}, \theta^{-2/3}]$, Algorithm $A_r^1$ attains the triple Pareto-optimal front for consistency, robustness, and smoothness among all deterministic algorithms for one-max-search. For $r \in [\theta^{-2/3}, \theta^{-1/2}]$, $A_r^1$ remains smooth and Pareto-optimal; however, its smoothness guarantee might admit further improvement.

We conclude this section with some observations. First, note that many learning-augmented algorithms in the literature express consistency and robustness in terms of a parameter $\lambda \in [0,1]$, which reflects the decision-maker's trust in the prediction. A simple yet effective parametrisation of $A_r^1$ can be achieved by setting $r = \theta^{-(1-\lambda/2)}$. Noting that $1/r\theta = \theta^{-\lambda/2}$, and $\frac{\ln\theta}{\ln(r\theta)} = \frac{2}{\lambda}$, the result of Theorem 3.2 can be restated, with this parametrization, as

$$\frac{A_r^1(p,y)}{p^*} \geq \max\left(\theta^{-(1-\frac{\lambda}{2})}, \theta^{-\frac{\lambda}{2}}\mathcal{E}(p^*,y)^{\max(1,\frac{2}{\lambda}-2)}\right) .$$

A second observation is that the obtained bounds can be readily adapted to the inverse ratio $p^*/A_r^1(p,y)$, which is also commonly used to define the competitive ratio in one-max-search (El-Yaniv, 1998). Specifically, by defining the inverse error as $\bar{\mathcal{E}} = 1/\mathcal{E} = \max\left\{\frac{p^*}{y}, \frac{y}{p^*}\right\}$, we obtain

$$\frac{p^*}{A_r^1(p,y)} \leq \min\left(\theta^{1-\frac{\lambda}{2}}, \theta^{\frac{\lambda}{2}}\bar{\mathcal{E}}(p^*,y)^{\max(1,\frac{2}{\lambda}-2)}\right) .$$

### 3.2. Extension to the additive error $\eta$

While the multiplicative error provides a more natural fit for the problem at hand, we also derive smoothness guarantees for $A_r^1$ using the additive error $\eta(p^*,y) = |p^* - y|$. Moreover, we prove that the smoothness it achieves is optimal for $\eta$, for all possible values of $r \in [\theta^{-1}, \theta^{-1/2}]$.

**Theorem 3.4.** *Let $A$ be any deterministic algorithm with robustness $r$ and consistency $1/r\theta$. Suppose that $A$ satisfies for all $p \in [1,\theta]^n$ and $y \in [1,\theta]$ that*

$$\frac{A(p,y)}{p^*} \geq \max\left(r, \frac{1}{r\theta} - \beta\frac{\eta(p^*,y)}{p^*}\right) \qquad (6)$$

*for some $\beta \geq 0$, then necessarily $\beta \geq \beta^*$, where*

$$\beta^* := \frac{1 - r^2\theta}{r\theta} \max\left(\frac{1}{1-r}, \frac{1}{r\theta - 1}\right) .$$

*Moreover, Algorithm $A_r^1$ satisfies (6) with $\beta = \beta^*$, which shows its optimality.*

The above theorem establishes that $A_r^1$ has the best possible smoothness guarantee amongst all Pareto-optimal algorithms. Consequently, it achieves a triple Pareto-optimal trade-off between consistency, robustness, and smoothness.

# 4. Stochastic One-Max Search

One-max-search under competitive analysis is a worst-case abstraction of online selection which is highly skewed towards pessimistic scenarios. This is an approach rooted in theoretical computer science that has the benefit of worst-case guarantees, but does not capture the stochasticity of real markets, e.g. (Cont & Tankov, 2004; Donnelly, 2022). In contrast, in mathematical (and practical) finance, probabilistic analyses such as risk management are preferred, e.g. (Merton, 1975). While reconciling the two approaches remains a very challenging perspective, we aim to narrow the very large gap between the worst-case and stochastic regimes by leveraging a probabilistic approach. This necessitates algorithms that can be robust to the randomness of the market, and to this end, the established smoothness of our algorithm (Section 3) will play a pivotal role, as we will show. A probabilistic analysis can thus yield two main practical benefits: 1) estimate performance under price distributions obtained from financial modelling; 2) leverage the consistency-robustness trade-off to handle risk.

In the stochastic formulation of one-max-search, we now consider the prices $(P_i)_{i=1}^n$ to be random variables whose maximum is $P^* \sim F^*$. Since market prices are random, the historical data used to generate a machine-learned prediction should also be random, hence we consider the prediction to be a random variable $Y \sim G$. As before, we consider that $P_i$, for $i \in [n]$, and $Y$ take value in $[1, \theta]$. The trading window unfolds as in the classic one-max-search problem, except that the prices and predictions are now random.

We will first give, in Section 4.1, a general probabilistic competitive analysis of the one-max-search problem which shows that the bounds of Section 3 transfer naturally by weighting the bounds of Theorem 3.2 according to the coupling of $(P^*, Y)$. In order to better understand the intuition behind these results, in Section 4.2, we instantiate the analysis with three insightful models. Finally, in Section 4.3, we analyse the effects of $F^*$ and $G$, first in isolation, and then jointly using analytical tools from optimal transport theory, to characterize how their interaction influences the outcome beyond their individual effects.

## 4.1. Competitive analysis in the stochastic framework

In the stochastic setting, we will evaluate the performance of the algorithm using the ratio of expectations $\mathbb{E}[\mathsf{ALG}(P^*, Y)]/\mathbb{E}[P^*]$, but our results and arguments transfer readily to $\mathbb{E}[\mathsf{ALG}(P^*, Y)/P^*]$.

Because any algorithm must operate on the realisation of $Y$, its performance becomes a random variable depending on the specific relationship of $P^*$ and $Y$. This is captured the

coupling $\pi^*$ of $(P^*, Y)$, yielding

$$\mathbb{E}[\mathsf{ALG}(P^*, Y)] := \int \mathsf{ALG}(p^*, y) \mathrm{d}\pi^*(p^*, y) . \quad (7)$$

In consequence, we can identify $\pi^*$ and the instance $(P^*, Y) \sim \pi^*$ without loss of generality, as all such instances are indistinguishable to a probabilistic analysis.

Taking into account the coupling, the bound of Theorem 3.4 adapts to the stochastic setting to yield Lemma 4.1.

**Lemma 4.1.** *The family* $\{\mathsf{A}_r^1\}_r$ *satisfies*

$$\frac{\mathbb{E}[\mathsf{A}_r^1(P^*, Y)]}{\mathbb{E}[P^*]} \geq \max\left\{r, \frac{1}{r\theta} \frac{\mathbb{E}\left[P^* \mathcal{E}(P^*, Y)^s\right]}{\mathbb{E}[P^*]}\right\} . \quad (8)$$

*Proof.* Apply Jensen's inequality to Theorem 3.2. $\square$

As expected, (8) shows that the robustness of $\{\mathsf{A}_r^1\}_r$ carries over to the stochastic setting through the $\max\{r, \cdot\}$ term.

## 4.2. Instantiations of Lemma 4.1

The coupling $\pi^*$, and Eq. (8) more broadly, encode effects that influence the quality of a prediction from two different sources: the relationship of $G$ and $F^*$ and the relationship between $Y$ and $P^*$ themselves (e.g. correlation). In this section, we aim to isolate the effect of $G$ and $F^*$.

**Stochastic predictions, deterministic prices**

This semi-deterministic model, in which $F^* = \delta_{p^*}$ (which is to say $P^* = p^*$ almost surely), isolates the effect of $G$. From a practical standpoint, it can also be used to model predictions which are noisy measurements of deterministic, but unknown, quantities. Its theoretical interest comes from the fact that it simplifies Eq. (7) into an integral over $F^*$. This allows us to derive Corollary 4.2 from Lemma 4.1, in which the function $\Lambda : [1, \theta] \to [0, 1]$ defined by

$$\Lambda(p^*) = \mathbb{E}\left[\mathcal{E}(P^*, Y)^s | P^* = p^*\right] \quad (9)$$

for $p^* \in [1, \theta]$, directly quantifies the quality of the prediction in terms of the performance, with respect to the true, realised, maximal price $p^*$. Indeed, $\Lambda(p^*) \leq 1$ for all $p^*$, and the closer to one, the better the prediction.

In particular, if the maximal price is deterministic, but the prediction is stochastic, this yields the following guarantees. Note that, for the sake of clarity of the results, we will no longer specify the term coming from the robustness, with the understanding that one can add a maximum with $r$ to any bound on the performance of $\{\mathsf{A}_r^1\}_r$.

**Corollary 4.2.** *Let* $F^* = \delta_{p^*}$ *for some* $p^* \in [1, \theta]$, *then the family* $\{\mathsf{A}_r^1\}_r$ *satisfies*

$$\frac{\mathbb{E}[\mathsf{A}_r^1(P^*, Y)]}{\mathbb{E}[P^*]} \geq \frac{1}{r\theta} \Lambda(p^*) . \quad (10)$$

Viewing $\Lambda$ as a map (taking $G$ to a real-valued function on $[1,\theta]$) reveals that it quantifies the usefulness of $G$ as a prediction distribution at any $p^* \in [1,\theta]$.

As an integral functional of $G$, $\Lambda$ may not admit a closed form. Nevertheless, it can be estimated to capture subtle stochastic phenomena as demonstrated by Example 4.3.

**Example 4.3.** *Let $F^* = \delta_{p^*}$ for some $p^* \in [1,\theta]$ and $G = \mathsf{Unif}([p^* - \epsilon, p^* + \epsilon])$. There is a constant $C > 0$, dependent only on $(s,\theta)$, such that*

$$\frac{\mathbb{E}[\mathsf{A}_r^1(P^*,Y)]}{\mathbb{E}[P^*]} \geq \frac{1}{r\theta}\left(1 - \frac{s}{2p^*}\epsilon - C\epsilon^2\right), \quad (11)$$

*as soon as $0 < \epsilon \leq \min\{\theta - p^*, p^* - 1\}$.*

Eq. (11) reveals that the performance of $\{\mathsf{A}_r^1\}_r$ decays from consistency at a rate linear in the uncertainty $\epsilon$, determined by the smoothness $s$ of the algorithm. This captures the scale of the effect of smoothness on a practical example. In Eq. (11) we characterised the rate up to the second order ($\epsilon^2$), but higher-order estimates can be obtained similarly.

Moreover, this shows that all sufficiently regular distributions can be approximated in terms of $\Lambda$ using mixtures over the model of Example 4.3, i.e. $G = \sum_{k=1}^{K} w_k \mathsf{Unif}(I_k)$ for $w_i > 0$, $\sum_i w_i = 1$, and $(I_k)_k$ disjoint subintervals of $[1,\theta]$ (see Corollary C.2). Numerical integration (e.g. Monte-Carlo) offers another alternative method to estimate $\Lambda$.

**Deterministic predictions, stochastic prices**

The performances of our family of algorithms can also be computed if the prices are stochastic, but the prediction is deterministic. This model swaps the randomness: now the prices are random so that $p^* \sim F^*$ is generic and it is $Y \sim \delta_y$ which is deterministic.

While this setting appears symmetrical to the previous one, this is not the case as the one-max-search problem itself is highly asymmetrical. Indeed, using a threshold means that predictions too high or too low do not have the same impact. By defining

$$\Upsilon(y) := \frac{\mathbb{E}[P^*\mathcal{E}(P^*,Y)^s | Y = y]}{\mathbb{E}[P^*]} \quad \text{for } y \in [1,\theta],$$

we can establish a quality quantification which mirrors $\Lambda$: this functional of $F^*$ states how good any unique prediction $y$ is at influencing algorithmic performance. This yields the following Corollary 4.4, an analogue of Corollary 4.2. Note that $\Upsilon(y) \leq 1$ for all $y \in [1,\theta]$.

**Corollary 4.4.** *Let $G = \delta_y$ for some $y \in [1,\theta]$. The family $\{\mathsf{A}_r^1\}_r$ satisfies*

$$\frac{\mathbb{E}[\mathsf{A}_r^1(P^*,Y)]}{\mathbb{E}[P^*]} \geq \frac{1}{r\theta}\Upsilon(y). \quad (12)$$

**Independent stochastic predictions and prices**

The theoretical value of the above two models is their isolation of the effect of $G$ into $\Lambda$ (resp. $F^*$ into $\Upsilon$). We now turn to a model in which $Y$ and $P^*$ are independent (denoted by $\pi^* = F^* \otimes G$) which will illustrate that predictions can be useful even without any correlation. The intuition is simple: some inaccurate predictions can still induce (on average) good thresholds because of the algorithm's internal mechanics. This effect is captured by the interaction between the functional $\Lambda$ and the distribution of prices $F^*$ (resp. $\Upsilon$ and $G$), as shown by Corollary 4.5[2].

**Corollary 4.5.** *Let $\pi^* = F^* \otimes G$, the family $\{\mathsf{A}_r^1\}_r$ satisfies*

$$\frac{\mathbb{E}[\mathsf{A}_r^1(P^*,Y)]}{\mathbb{E}[P^*]} \geq \frac{1}{r\theta}\int \Upsilon(y)\mathrm{d}G(y)$$

$$= \frac{1}{r\theta}\int \frac{p^*\Lambda(p^*)}{\mathbb{E}[P^*]}\mathrm{d}F^*(p^*). \quad (13)$$

Since $\Lambda(z)$ is always smaller than 1 (and again, the closer to one, the better the predictions are), Eq. (13) gives an intuitive bound on the performance of the algorithms.

The theoretical benefit of the model transpires in Corollary 4.5: independence separates the integral against $\pi^*$ in Eq. (7) into a double integral revealing $\Upsilon$. Unfortunately, it is often difficult to obtain a closed form for the resulting expression (see, e.g. , Proposition C.5), but one can rely on numerical integration instead (see Figure 5 in Appendix C).

## 4.3. Dependent predictions and optimal transport

The previous models successfully isolated the effect of the distributions $F^*$ and $G$. Using tools from Optimal Transport (OT) theory, one can generalise this approach. For brevity, we refer simply to (Villani, 2009) for the technicalities and background of this field. The key observation is that the right-hand side of Eq. (7) is a *transport functional* of $\pi^*$, which can be lower bounded uniformly over the set of couplings $\Pi(F^*, G)$ of $F^*$ and $G$. This set is exactly the set of joint distributions for $(P^*, Y)$ when $P^* \sim F^*$ and $Y \sim G$. Minimising a transport functional over couplings is the classic OT problem (Villani, 2009), hence Theorem 4.6.

**Theorem 4.6.** *The family $\{\mathsf{A}_r^1\}_r$ satisfies*

$$\frac{\mathbb{E}[\mathsf{A}_r^1(P^*,Y)]}{\mathbb{E}[P^*]} \geq \frac{1}{r\theta}\frac{\inf \int p^*\mathcal{E}(p^*,y)^s\mathrm{d}\pi(p^*,y)}{\mathbb{E}[P^*]}. \quad (14)$$

*where the infimum is taken over couplings $\pi \in \Pi(F^*, G)$ ; in particular, this implies the numerator is as most $\mathbb{E}[P^*]$.*

Theorem 4.6 highlights a novel connection between (stochastic) competitive analysis and optimal transport. Con-

---

[2]The following result also applies to sampling of distribution-valued predictions (Angelopoulos et al., 2024; Dinitz et al., 2024).

trary to most literature in OT, in which the optimal configuration tries to minimise the distance points $(p^*, y)$ are moved, the infimum in Eq. (14) tries to push them far apart to induce the algorithm to make mistakes.

Optimal transport tools have been used before in algorithms with predictions, notably in the *distributional predictions* setting in which the algorithm is given $G$ itself (Angelopoulos et al., 2024; Dinitz et al., 2024). This analysis, however, is fundamentally different: it uses Wasserstein distances (see Appendix D.4) in place of $\eta$ in quantifying the error of the distributional prediction $G$ of $F^*$. Our stochastic framework ties its error metric closely to the asymmetric nature of the problem through $p^*\mathcal{E}(p^*, y)^s$, which is why our OT problem, i.e. Eq. (14), is *not* symmetric: exchanging the roles of $(G, F^*)$ cannot be expected to yield the same performance.

The optimal transport problem in Eq. (14) generally has no closed form, but thanks to its (strong) dual form (see Appendix C.2), one can use problem-specific knowledge to derive lower bounds, as demonstrated by Proposition 4.7.

**Proposition 4.7.** *The family $\{\mathsf{A}_r^1\}_r$ satisfies*

$$\frac{\mathbb{E}[\mathsf{A}_r^1(P^*, Y)]}{\mathbb{E}[P^*]} \geq \frac{1}{r\theta} \frac{\int_1^{\theta^{\frac{1}{2}}} p^{*1+s}\mathrm{d}F^*(p^*) + \int_{\theta^{\frac{1}{2}}}^{\theta} p^{*1-s}\mathrm{d}F^*(p^*)}{\mathbb{E}[P^*]}.$$

*Moreover, the RHS is the infimum over $G$ of Eq.* (14).

Proposition 4.7 once more highlights the asymmetry of the problem through different contributions of the regions above and below $\sqrt{\theta}$, which is the threshold that guarantees $1/\sqrt{\theta}$-robustness. The dual problem provides thus a practical tool for designing lower bounds for the performance of $\{\mathsf{A}_r^1\}_r$ in the stochastic one-max-search setting.

# 5. Numerical Experiments

To complement our theoretical analysis and evaluate the performance of our algorithm in practice, we present experimental results in this section. We defer additional experimental results to Appendix E.

## 5.1. Experiments on synthetic data

We fix $\theta = 5$, $\lambda = 0.5$, and $r = \theta^{-(1-\lambda/2)}$. We consider instances $\{\mathcal{I}_n(q)\}_{q \in [1,\theta]}$, where $\mathcal{I}_n(q)$ is the sequence starting at 1, and increasing by $\frac{\theta-1}{n-1}$ at each step until reaching $q$, after which the prices drop to 1. A more formal definition of this instance can be found in Equation 15. These instances model worst-case instances with maximum price $q$, and they are used in general to prove impossibility results in the one-max-search problem.

We fix an error level $\mathcal{E}_{\min}$ and, for each $p^* \in [1, \theta]$, we

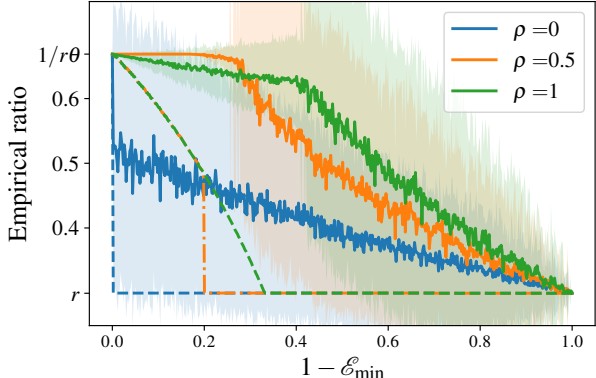

*Figure 3.* Performance of $\mathsf{A}_r^\rho$ with $\rho \in \{0, 0.5, 1\}$.

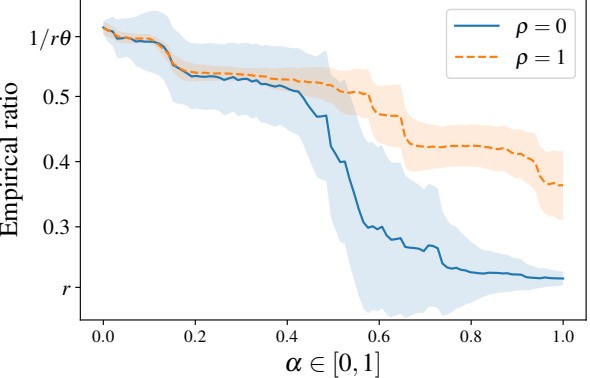

*Figure 4.* Comparison of $\mathsf{A}_r^1$ and $\mathsf{A}_r^0$ on the Bitcoin price dataset.

generate the prediction $y$ by sampling uniformly at random in the interval $[p^*\mathcal{E}_{\min}, p^*/\mathcal{E}_{\min}] = \{z : \mathcal{E}(p^*, z) \geq \mathcal{E}_{\min}\}$, then compute the ratio $\mathsf{A}_r^\rho(\mathcal{I}_n(p^*), y)/p^*$. For $\mathcal{E}_{\min} \in (0, 1]$, Figure 3 illustrates the worst-case ratio $\inf_{p^* \in [1,\theta]} \mathbb{E}_y[\mathsf{A}_r^\rho(p^*, y)]/p^*$, where the expectation is estimated empirically using 500 independent trials. The figure also shows, with dashed lines, the theoretical worst-case ratio corresponding to the given error level $\mathcal{E}_{\min}$.

Figure 3 shows that for the different values of $\rho$, the worst-case ratio is $1/r\theta$ when the prediction is perfect, i.e. $\mathcal{E}_{\min} = 1$, and degrades to $r$ when the prediction can be arbitrarily bad, which is consistent with Theorem 3.1. However, the ratio achieved for $\rho = 0$ drops significantly even with a slight perturbation in the prediction, while the ratios with $\rho \in \{0.5, 1\}$ decrease much slower. This is again consistent with the smoothness of $\mathsf{A}_r^\rho$, as shown in Theorem 3.2.

## 5.2. Experiments on real data

To further validate our algorithm's practicality, we evaluate it on the experimental setting of (Sun et al., 2021). Specifi-

cally, we use real Bitcoin data (USD) recorded every minute from the beginning of 2020 to the end of 2024. The dataset's prices range from $L = 3,858$ USD to $U = 108,946$ USD, yielding $\theta = U/L \approx 28$.

We randomly sample a 10-week window of prices $W_0$, let $W_{-1}$ be the 10-week window of prices preceding $W_0$ and take the prediction $y = \alpha \max W_{-1} + (1 - \alpha) \max W_0$. Here, $\alpha$ captures the prediction error: $\alpha = 0$ represents perfect foresight, while $\alpha = 1$ corresponds to a naive prediction equal to the maximum price in $W_{-1}$. To simulate worst-case scenarios, the last price in $W_0$ is changed to $L$ with a probability of 0.75. For each value of $\alpha$, we sample $m = 100$ windows $(W_0^j)_{j=1}^m$ and compute the ratio $R_m = \min_j \{\mathsf{A}_r^\rho(W_0^j, y) / \max W_0^j\}$ for $\rho \in \{0, 1\}$. We then empirically estimate $\mathbb{E}[R_m]$ by repeating this process 50 times. We choose the robustness $r$ of $\mathsf{A}_r^\rho$ by setting $\lambda \in [0, 1]$ and $r = \theta^{-(1-\lambda/2)}$.

Figure 4 shows the obtained results with $\lambda = 0.5$, and compares our algorithm $\mathsf{A}_r^1$ to the algorithm of (Sun et al., 2021), which corresponds to $\mathsf{A}_r^0$. For $\alpha = 0$, i.e. perfect prediction, they both achieve the consistency $1/r\theta$. However, as the error increases, $\mathsf{A}_r^0$ quickly degrades to the robustness guarantee $r$, whereas $\mathsf{A}_r^1$ degrades more gradually.

## 6. Conclusion

We provided an intuitive Pareto-optimal and smooth algorithm for a fundamental online decision problem, namely one-max-search. We believe our methodology can be applied to generalizations such as the *k-search* problem (Lorenz et al., 2009), i.e. multi-unit one-max-search, recently studied in a learning-augmented setting (Lee et al., 2024). More broadly, we believe our framework can help bring competitive analysis much closer to the analysis of real financial markets since it combines three essential aspects: worst-case analysis, adaptivity to stochastic settings, and smooth performance relative to the error. A broader research direction is thus to extend the study of competitive financial optimization (see, e.g. , Chapter 14 in (Borodin & El-Yaniv, 2005)) to such realistic learning-augmented settings. This work also sheds light on connections between competitive analysis and optimal transport, suggesting the study of the geometry of OT problems induced by competitive analysis as a promising direction for both theories.

Our work also lays the groundwork for exploring triple Pareto-optimal frontiers among consistency, robustness, and smoothness. While our results establish optimal worst-case smoothness guarantees for both multiplicative and additive prediction errors, an intriguing direction for future research is to investigate how these bounds improve under additional assumptions, such as when the prediction error is known to be bounded.

## Impact Statement

This paper presents a work whose goal is to advance the field of learning-augmented algorithms. There are many potential societal consequences of our work, none of which we feel must be specifically highlighted here.

## Acknowledgements

This research was supported in part by the French National Research Agency (ANR) in the framework of the PEPR IA FOUNDRY project (ANR-23-PEIA-0003) and through the grant DOOM ANR23-CE23-0002. It was also funded by the European Union (ERC, Ocean, 101071601). Views and opinions expressed are however those of the author(s) only and do not necessarily reflect those of the European Union or the European Research Council Executive Agency. Neither the European Union nor the granting authority can be held responsible for them.

This work was also partially funded by the project PRE-DICTIONS, grant ANR-23-CE48-0010 from the French National Research Agency (ANR).

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

# Appendices

## A. Organisation and Notation

### A.1. Organisation of Appendices

The following appendices are divided in the following way: Appendix B contains the proofs of Section 3, while Appendix C contains proofs of Section 4. In both cases, they follow the order of the main text. Appendix E provides further experiments not included in Section 5. After these sections which are ordered according to the text, Appendix D is transversal and regroups results in which the error analysis is additive instead of multiplicative.

### A.2. Notation

The space of probability measures over a set $S$ is denoted $\mathscr{P}(S)$. The set of couplings between $G$ and $F^*$ is $\Pi(G, F^*) := \{\pi \in \mathscr{P}([1,\theta]^2) : \pi(\cdot, [1,\theta]) = G, \pi([1,\theta], \cdot) = F^*\}$ For $x \in \mathbb{R}$, $\delta_x$ denotes the Dirac Delta distribution (i.e. the distribution of a degenerate random variable $X$ satisfying $\mathbb{P}(X = x) = 1$).

## B. Proofs of Section 3

We present in this appendix the proofs of the results stated in Section 3. Let us introduce some notations and observations that we will use throughout the proofs. For all $n \geq 1$, we denote by $(p_i^n)_{i=1}^n$ the sequence of prices defined by

$$\forall i \in [n]: \quad p_i^n = 1 + \frac{\theta - 1}{n - 1}(i - 1),$$

and for all $q \in [1, \theta]$, we denote by $\mathcal{I}_n(q)$ the sequence of prices that are equal to $p_i^n$ while the latter is smaller than $q$ then drops to 1. Formally, the $i^{\text{th}}$ price in this sequence is

$$\mathcal{I}_n(q)_i = 1 + (p_i^n - 1)\mathbf{1}_{p_i^n \leq q}. \tag{15}$$

On the class of instances $\{\mathcal{I}_n(q)\}_{q \in [1,\theta]}$, any deterministic learning-augmented algorithm for one-max-search is equivalent to a single-threshold algorithm $\mathsf{A}_\Phi$. Moreover, observe that the payoff of $\mathsf{A}_\Phi$ satisfies

$$\mathsf{A}_\Phi(\mathcal{I}_n(q), y) = \begin{cases} 1 & \text{if } \Phi(y) > q \\ \Phi(y) + O(\frac{1}{n}) & \text{otherwise} \end{cases}. \tag{16}$$

Indeed, if the threshold exceeds the maximum price $q$ in the sequence, then no price is selected and the algorithm is left with a payoff of 1. On the hand, if the threshold is at most $q$, then the selected price is $\min\{p_i^n \mid p_i^n \geq \Phi(y)\}$. By definition of the prices $p_i^n$, this value is in $[\Phi(y), \Phi(y) + \frac{\theta-1}{n-1}]$. In particular, for $n$ arbitrarily large, the payoff of an algorithm with threshold $\Phi(y) \leq q$ is arbitrarily close $\Phi(y)$. This observation will be useful in our proofs.

### B.1. The class of all Pareto-optimal thresholds

**Theorem 3.1.** *For any fixed of robustness $r$, the set of all thresholds $\Phi : [1, \theta] \to [1, \theta]$ such that $\mathsf{A}_\Phi$ has robustness $r$ and consistency $1/r\theta$ is*

$$\mathcal{P}_r := \Big\{\Phi \; : \forall z \in [1, \theta] : r\theta \leq \Phi(z) \leq \frac{1}{r}$$
$$\forall z \in [r\theta, \theta] : \frac{z}{r\theta} \leq \Phi(z) \leq z\Big\}.$$

*Proof.* Let $\Phi : [1, \theta] \to [1, \theta]$, and consider that $\mathsf{A}_\Phi$ is $r$-robust and $1/r\theta$-consistent. We will prove that $\Phi$ is necessarily in the set $\mathcal{P}_r$.

**First inclusion.** Let us first prove that $\Phi(z) \in [r\theta, 1/r]$ for all $z \in [1, \theta]$. Let $z \in [1, \theta]$, and consider the sequence of prices $\mathcal{I}_n(q)$. Since $\mathsf{A}_\Phi$ is $r$-robust, then by Eq. (16), it holds for $q = \Phi(z) - \frac{1}{n}$ and $y = z$ that

$$r \leq \frac{\mathsf{A}_\Phi(\mathcal{I}_n(q), z)}{q} = \frac{1}{q} = \frac{1}{\Phi(z) - \frac{1}{n}}.$$

On the other hand, for $q = \theta$, we obtain again using Eq. (16) that

$$r \leq \frac{\mathsf{A}_\Phi(\mathcal{I}_n(q), z)}{q} = \frac{\Phi(z) + O(\frac{1}{n})}{\theta} \; .$$

We deduce that $\Phi(z)$ satisfies $\Phi(z) \in [r\theta - O(\frac{1}{n}), 1/r + \frac{1}{n}]$, and taking the limit for $n \to \infty$ gives that $\Phi(z) \in [r\theta, 1/r]$.

Consider now $z \in (r\theta, \theta]$, and let us prove that $\Phi(z) \in [\frac{z}{r\theta}, z]$. We first prove by contradiction that $\Phi(z) \leq z$. Suppose this is not the case, i.e. $\Phi(z) > z$ then using Equation (16) and that the algorithm is $1/r\theta$-consistent, considering that the maximum price is $z$ and the prediction is perfect, we deduce that

$$\frac{1}{r\theta} \leq \frac{\mathsf{A}_\Phi(\mathcal{I}_n(z), z)}{z} = \frac{1}{z} \; ,$$

hence $z \leq r\theta$, which contradicts the initial assumption that $z \in (r\theta, \theta]$. Therefore, $\Phi(z) \leq z$ for all $z \in (r\theta, \theta]$. Using this inequality, it follows again by Eq. (16) and $1/r$-consistency of the algorithm that

$$\frac{1}{r\theta} \leq \frac{\mathsf{A}_\Phi(\mathcal{I}_n(z), z)}{z} = \frac{\Phi(z)}{z} \; , .$$

Consequently, $\Phi(z) \in [\frac{z}{r\theta}, z]$ for all $z \in (r\theta, \theta]$. This proves that the set of all thresholds yielding Pareto-optimal levels of robustness and consistency $(r, 1/r\theta)$ is included in the set $\mathcal{P}_r$.

**Second inclusion.** The other inclusion is easier to prove. Let $\Phi \in \mathcal{P}_r$, and let us prove that $\mathsf{A}_\Phi$ is $r$-robust and $1/r\theta$-consistent. Consider an arbitrary sequence of prices $p = (p_i)_{i=1}^n \in [1, \theta]$ and a prediction $y \in [1, \theta]$, and let us denote by $p^*$ the maximum price in the sequence $p$.

The robustness of $\mathsf{A}_\Phi$ follows from the bounding $\Phi(y) \in [r\theta, 1/r]$. Indeed, we obtain using Eq. (16) that

- if $p^* < \Phi(y)$ then $\frac{\mathsf{A}_\Phi(p,y)}{p^*} = \frac{1}{p^*} \geq \frac{1}{\Phi(y)} \geq r$,

- if $p^* \geq \Phi(y)$ then $\frac{\mathsf{A}_\Phi(p,y)}{p^*} = \frac{\Phi(y)}{p^*} \geq \frac{\Phi(y)}{\theta} \geq r$ ,

which proves that $\mathsf{A}_\Phi$ is $r$-robust. Now the consistency of the algorithm follows from the bounding $\Phi(z) \in [\frac{z}{r\theta}, z]$ for all $z \in (r\theta, \theta]$. Indeed, assume that the prediction is perfect, i.e. $y = p^*$, then we have the following:

- if $p^* \leq r\theta$ then $\frac{\mathsf{A}_\Phi(p,p^*)}{p^*} = \frac{1}{p^*} \geq \frac{1}{r\theta}$,

- if $p^* > r\theta$ then we have that $\Phi(p^*) \leq p^*$, hence $\frac{\mathsf{A}_\Phi(p,p^*)}{p^*} = \frac{\Phi(p^*)}{p^*} \geq \frac{1}{r\theta}$.

This proves $\mathsf{A}_\Phi$ is $1/r\theta$-consistent, which concludes the proof. $\qquad\square$

## B.2. Smoothness analysis of $\mathsf{A}_r^\rho$

**Lemma B.1.** *Let $a > 0$, $b \in \mathbb{R}$, and $u < v \in (0, \infty)$ satisfying that $z \mapsto az + b \geq 0$ on the interval $[u,v]$, then it holds for all $\ell \in \mathbb{R}$ that*

$$\max_{z \in [u,v]} \frac{(az + b)^{\ell+1}}{z^\ell} = \max\left\{ \frac{(au + b)^{\ell+1}}{u^\ell}, \frac{(av + b)^{\ell+1}}{v^\ell} \right\} \; .$$

*Proof.* For all $z \in [u, v]$, we can write that

$$\frac{(az + b)^{\ell+1}}{z^\ell} = \left( \frac{az + b}{z^{\ell/(\ell+1)}} \right)^{\ell+1} = \left( az^{1/(\ell+1)} + bz^{-\ell/(\ell+1)} \right)^{\ell+1}$$

hence, computing the derivative gives

$$
\frac{d}{dz}\left[\frac{(az+b)^{\ell+1}}{z^\ell}\right] = \frac{d}{dz}\left[\left(az^{1/(\ell+1)} + bz^{-\ell/(\ell+1)}\right)^{\ell+1}\right]
$$

$$
= (\ell+1)\left(\frac{a}{\ell+1}z^{-\ell/(\ell+1)} - \frac{b\ell}{\ell+1}z^{-\ell/(\ell+1)-1}\right)\left(az^{1/(\ell+1)} + bz^{-\ell/(\ell+1)}\right)^\ell
$$

$$
= az^{-\ell/(\ell+1)-1}\left(z - \frac{b\ell}{a}\right)\left(az^{1/(\ell+1)} + bz^{-\ell/(\ell+1)}\right)^\ell .
$$

The monotonicity of $z \mapsto \frac{(az+b)^{\ell+1}}{z^\ell}$ on the interval $[u,v]$ is therefore determined by the sign of $z - \frac{b\ell}{a}$. Indeed, $az^{-\ell/(\ell+1)-1} \geq 0$ because $a \geq 0$ and $z \geq u > 0$, and the term $\left(az^{1/(\ell+1)} + bz^{-\ell/(\ell+1)}\right)^\ell$ is also non-negative because we can write that

$$
\left(az^{1/(\ell+1)} + bz^{-\ell/(\ell+1)}\right)^\ell = \left[\left(az^{1/(\ell+1)} + bz^{-\ell/(\ell+1)}\right)^{\ell+1}\right]^{\frac{\ell}{\ell+1}} = \left[\frac{(az+b)^{\ell+1}}{z^\ell}\right]^{\frac{\ell}{\ell+1}} ,
$$

and both $z$ and $az + b$ are positive on the interval $[u,v]$.

Consequently, depending on how $\frac{b\ell}{a}$ compares to the bounds $u, v$ of the interval, the mapping $z \mapsto \frac{(az+b)^{\ell+1}}{z^\ell}$ can be either decreasing, increasing, or decreasing then increasing. In the three cases, its maximum is reached in one of the interval limits $u$ or $v$, thus

$$
\max_{z\in[u,v]} \frac{(az+b)^{\ell+1}}{z^\ell} = \max\left\{\frac{(au+b)^{\ell+1}}{u^\ell}, \frac{(av+b)^{\ell+1}}{v^\ell}\right\} .
$$

$\square$

**Corollary B.2.** *For all $\rho \in (0,1]$, for $s = \frac{1}{\rho}\left(\frac{\ln\theta}{\ln(r\theta)} - 2\right)$ it holds that*

$$
\max_{z\in[\frac{1}{r}, \frac{1}{r}+\rho(\theta-\frac{1}{r})]} \frac{\Phi_r^\rho(z)^{s+1}}{z^s} \leq r\theta .
$$

*Proof.* By definition of the threshold $\Phi_r^\rho$, we have for $z \in [\frac{1}{r}, \frac{1}{r} + \rho(\theta - \frac{1}{r})]$

$$
\Phi_r^\rho(z) = \varphi_r(z) + \frac{\frac{1}{r} - \varphi_r(z)}{\rho(\theta - \frac{1}{r})}\left(z - \frac{1}{r}\right) ,
$$

which can be written as $az + b$ with $a = \frac{1/r - \varphi_r(z)}{\rho(\theta - 1/r)} \geq 0$ because $\varphi_r(z) < 1/r$ for all $z \leq \theta$. Consequently, Lemma B.1 gives that

$$
\max_{z\in[\frac{1}{r}, \frac{1}{r}+\rho(\theta-\frac{1}{r})]} \frac{\Phi_r^\rho(z)^{s+1}}{z^s} = \max\left\{\frac{\Phi_r^\rho(1/r)^{s+1}}{1/r^s}, \frac{\Phi_r^\rho(\frac{1}{r} + \rho(\theta - \frac{1}{r}))^{s+1}}{(\frac{1}{r} + \rho(\theta - \frac{1}{r}))^s}\right\}
$$

$$
= \max\left\{\frac{\varphi_r(1/r)^{s+1}}{1/r^s}, \frac{1/r^{s+1}}{(\frac{1}{r} + \rho(\theta - \frac{1}{r}))^s}\right\} . \tag{17}
$$

We will now prove that both terms in the maximum are at most equal to $r\theta$.

For all $h \in \mathbb{R}$, we have the equivalences

$$
\frac{1/r^{h+1}}{(\frac{1}{r} + \rho(\theta - \frac{1}{r}))^h} \leq r\theta \iff \frac{1}{(1 + \rho(r\theta - 1))^h} \leq r^2\theta
$$

$$
\iff -h\ln\left(1 + \rho(r\theta - 1)\right) \leq \ln(r^2\theta) = -\left(\ln\theta - 2\ln(r\theta)\right)
$$

$$
\iff h \geq \frac{\ln\theta - 2\ln(r\theta)}{\ln\left(1 + \rho(r\theta - 1)\right)} .
$$

Moreover, we have by concavity of $x \mapsto \ln x$ that

$$\ln \left( 1 + \rho(r\theta - 1) \right) = \ln \left( \rho r\theta + (1 - \rho) \right) \geq \rho \ln(r\theta) + (1 - \rho) \ln 1 = \rho \ln(r\theta) \,,$$

therefore, $s = \frac{1}{\rho} \left( \frac{\ln \theta}{\ln(r\theta)} - 2 \right)$ satisfies

$$s = \frac{1}{\rho} \left( \frac{\ln \theta}{\ln(r\theta)} - 2 \right) = \frac{\ln \theta - 2 \ln(r\theta)}{\rho \ln(r\theta)} \geq \frac{\ln \theta - 2 \ln(r\theta)}{\ln \left( 1 + \rho(r\theta - 1) \right)} \,,$$

and we deduce with the previous equivalences that

$$\frac{1/r^{s+1}}{\left( \frac{1}{r} + \rho(\theta - \frac{1}{r}) \right)^s} \leq r\theta \,. \tag{18}$$

Let us now prove that $\frac{\varphi_r(1/r)^{s+1}}{1/r^s} \leq r\theta$. Since $\varphi_r$ is a linear mapping with a positive slope, using that $1/r \in [r\theta, \theta]$ and Lemma B.1, we obtain that

$$\begin{aligned}
\frac{\varphi_r(1/r)^{s+1}}{1/r^s} &\leq \max_{z \in [r\theta, \theta]} \frac{\varphi_r(z)^{s+1}}{z^s} \\
&= \max \left\{ \frac{\varphi_r(r\theta)^{s+1}}{(r\theta)^s}, \frac{\varphi_r(\theta)^{s+1}}{\theta^s} \right\} \\
&= \max \left\{ \frac{(r\theta)^{s+1}}{(r\theta)^s}, \frac{(1/r)^{s+1}}{\theta^s} \right\} \\
&= \max \left\{ r\theta, \frac{\theta}{(r\theta)^{s+1}} \right\} \,.
\end{aligned}$$

Observing that $k = \rho s = \frac{\ln \theta}{\ln r\theta} - 2$ is the solution of $\frac{\theta}{(r\theta)^{k+1}} = r\theta$, and given that $k \leq s$ and $\rho \leq 1$, we have that

$$\frac{\theta}{(r\theta)^{s+1}} \leq \frac{\theta}{(r\theta)^{k+1}} = r\theta \,,$$

hence

$$\frac{\varphi_r(1/r)^{s+1}}{1/r^s} \leq \frac{\varphi_r(1/r)^{k+1}}{1/r^k} = \max \left\{ r\theta, \frac{\theta}{(r\theta)^{k+1}} \right\} = r\theta \,. \tag{19}$$

Finally, using Equations (17), (18), and (19), we deduce that

$$\max_{z \in [\frac{1}{r}, \frac{1}{r} + \rho(\theta - \frac{1}{r})]} \frac{\Phi_r^\rho(z)^{s+1}}{z^s} \leq r\theta \,,$$

which concludes the proof. $\qquad\square$

**Theorem 3.2.** *The family* $\{\mathsf{A}_r^\rho\}_{r,\rho}$ *satisfies*

$$\frac{\mathsf{A}_r^\rho(p, y)}{p^*} \geq \max \left( r, \frac{1}{r\theta} \mathcal{E}(p^*, y)^{s_\rho} \right) \,, \tag{4}$$

*with* $s_\rho := \max \left( 1, \frac{1}{\rho} \left( \frac{\ln \theta}{\ln(r\theta)} - 2 \right) \right)$, *for* $\rho \in [0, 1]$.

*Proof.* Consider $r \in [\theta^{-1}, \theta^{-1/2}]$, an instance $p = (p_i)_{i=1}^n$, a prediction $y$ of $p^*$, and let $\mathcal{E} = \min(\frac{y}{p^*}, \frac{p^*}{y})$. We will prove the smoothness guarantee separately on the intervals $[1, r\theta)$, $[r\theta, 1/r)$, $[1/r, 1/r + \rho(\theta - 1/r)]$, and $[1/r + \rho(\theta - 1/r), \theta]$. To lighten the notation, we simply write $s$ for $\max \left( 1, \frac{1}{\rho} \left( \frac{\ln \theta}{\ln(r\theta)} - 2 \right) \right)$ instead of $s_\rho$.

**Case 1.** If $y \in [1, r\theta)$, then $\Phi_r^\rho(y) = r\theta$. If $p^* < r\theta$ then

$$\frac{\mathsf{A}_r^\rho(p, y)}{p^*} \geq \frac{1}{p^*} \geq \frac{1}{r\theta} \geq \frac{1}{r\theta}\mathcal{E}^s ,$$

and if $p^* \geq r\theta$ then the payoff of the algorithm is at least equal to the threshold $r\theta$, hence

$$\frac{\mathsf{A}_r^\rho(p, y)}{p^*} \geq \frac{\Phi_r^\rho(y)}{p^*} = \frac{r\theta}{p^*} \geq \frac{y}{p^*} \geq \frac{1}{r\theta}\mathcal{E} \geq \frac{1}{r\theta}\mathcal{E}^s .$$

We used in the last two inequalities that $\frac{y}{p^*} \geq \mathcal{E}$, $r\theta \geq 1$ and $s \geq 1$.

**Case 2.** Consider now the case of $y \in [r\theta, 1/r]$, then $\Phi_r^\rho(y) = \varphi_r(y) = \frac{r\theta - 1}{1 - r} + \frac{1 - r^2\theta}{1 - r} \cdot \frac{y}{r\theta}$. If $p^* \geq \Phi_r^\rho(y)$ then the payoff of the algorithm is at least equal to the threshold. Using that $\frac{y}{p^*} \geq \mathcal{E}$, $1 \geq \mathcal{E}$ and $p^* \leq \theta$, we obtain

$$
\begin{aligned}
\frac{\mathsf{A}_r^\rho(p, y)}{p^*} &\geq \frac{\Phi_r^\rho(y)}{p^*} = \frac{\varphi_r(y)}{p^*} \\
&= \frac{r\theta - 1}{1 - r} \cdot \frac{1}{p^*} + \frac{1 - r^2\theta}{1 - r} \cdot \frac{1}{r\theta} \cdot \frac{y}{p^*} \\
&\geq \left( \frac{r\theta - 1}{1 - r} \cdot \frac{1}{p^*} + \frac{1 - r^2\theta}{1 - r} \cdot \frac{1}{r\theta} \right) \mathcal{E} \\
&\geq \left( \frac{r\theta - 1}{1 - r} \cdot \frac{1}{\theta} + \frac{1 - r^2\theta}{1 - r} \cdot \frac{1}{r\theta} \right) \mathcal{E} \\
&= \frac{1}{r\theta(1 - r)} \left( r^2\theta - r + 1 - r^2\theta \right) \mathcal{E} \\
&= \frac{1}{r\theta}\mathcal{E} \geq \frac{1}{r\theta}\mathcal{E}^s .
\end{aligned}
\tag{20}
$$

On the other hand, if $p^* < \Phi_r^1(y)$, recalling that $\Phi_r^\rho(z) = \varphi_r(z)$ for $z \in [r\theta, 1/r]$, we can use Inequality (19) from the proof of Corollary B.2, which gives for $k = \frac{\ln\theta}{\ln(r\theta)} - 2$ that

$$\max_{z \in [r\theta, \theta]} \frac{\Phi_r^\rho(z)^{k+1}}{z^k} = \frac{\varphi_r(z)^{k+1}}{z^k} = \max\left\{ r\theta, \frac{\theta}{(r\theta)^{k+1}} \right\} = r\theta ,$$

and it follows that

$$
\begin{aligned}
\frac{\mathsf{A}_r^\rho(p, y)}{p^*} &\geq \frac{1}{p^*} \geq \frac{1}{\Phi_r^\rho(y)} \\
&\geq \frac{1}{\Phi_r^\rho(y)} \cdot \frac{1}{r\theta} \cdot \frac{\Phi_r^\rho(y)^{k+1}}{y^k} = \frac{1}{r\theta} \left( \frac{\Phi_r^\rho(y)}{y} \right)^k \\
&\geq \frac{1}{r\theta} \left( \frac{p^*}{y} \right)^k \geq \frac{1}{r\theta}\mathcal{E}^k \geq \frac{1}{r\theta}\mathcal{E}^s ,
\end{aligned}
$$

where we used in the last inequality that $\mathcal{E} \leq 1$ and $k \leq \frac{k}{\rho} \leq \max(1, \frac{k}{\rho}) = s$.

**Case 3.** For $y \in [1/r, 1/r + \rho(\theta - 1/r)]$, if $p^* \geq \Phi_r^\rho(y)$, then observing that $\Phi_r^\rho(y) \geq \varphi_r(y)$, we obtain with the same computation as Eq. (20) that

$$\frac{\mathsf{A}_r^\rho}{p^*} \geq \frac{\Phi_r^\rho(y)}{p^*} \geq \frac{\varphi_r(y)}{p^*} \geq \frac{1}{r\theta}\mathcal{E}^s .$$

On the other hand, if $p^* < \Phi_r^\rho(y)$, then by Corollary B.2, we have for $k = \frac{\ln\theta}{\ln(r\theta)} - 2$ that

$$\max_{z \in [\frac{1}{r}, \frac{1}{r} + \rho(\theta - \frac{1}{r})]} \frac{\Phi_r^\rho(z)^{\frac{k}{\rho} + 1}}{z^{\frac{k}{\rho}}} \leq r\theta ,$$

therefore, the ratio between the algorithm's payoff and the maximum payoff can be lower bounded as follows

$$\frac{\mathsf{A}_r^\rho(p,y)}{p^*} \geq \frac{1}{p^*} \geq \frac{1}{\Phi_r^\rho(y)}$$

$$\geq \frac{1}{\Phi_r^\rho(y)} \cdot \frac{1}{r\theta} \cdot \frac{\Phi_r^\rho(y)^{\frac{k}{\rho}+1}}{y^{\frac{k}{\rho}}} = \frac{1}{r\theta}\left(\frac{\Phi_r^\rho(y)}{y}\right)^k$$

$$\geq \frac{1}{r\theta}\left(\frac{p^*}{y}\right)^{\frac{k}{\rho}} \geq \frac{1}{r\theta}\mathcal{E}^{\frac{k}{\rho}} \geq \frac{1}{r\theta}\mathcal{E}^s \ .$$

The last inequality holds because $\frac{k}{\rho} \leq \max(1, \frac{k}{\rho}) = s$.

**Case 4.** Finally, if $y \in (1/r + \rho(\theta - 1/r), \theta]$, then $\Phi_r^\rho(y) = 1/r$. If $p^* \geq 1/r$, then we have immediately that

$$\frac{\mathsf{A}_r^\rho(p,y)}{p} \geq \frac{\Phi_r^\rho(y)}{p^*} = \frac{1}{rp^*} \geq \frac{1}{r\theta} \geq \frac{1}{r\theta}\mathcal{E}^s \ .$$

Now if $p^* < 1/y$, let $k = \frac{\ln\theta}{\ln(r\theta)} - 2$ and $z_\rho = 1/r + \rho(\theta - 1/r)$. Observing that $s \geq \frac{k}{\rho} \geq k$, and $y \geq z_\rho$, and $ry \geq 1$, we deduce from Corollary B.2

$$\frac{1}{r}\left(\frac{1}{ry}\right)^s \leq \frac{1}{r}\left(\frac{1}{ry}\right)^{k/\rho} = \frac{(1/r)^{\frac{k}{\rho}+1}}{y^{k/\rho}} \leq \frac{(1/r)^{\frac{k}{\rho}+1}}{z_\rho^{k/\rho}} = \frac{\Phi(z_\rho)^{\frac{k}{\rho}+1}}{z_\rho^{k/\rho}} \leq \max_{z\in[\frac{1}{r},z_\rho]} \frac{\Phi_r^\rho(z)^{\frac{k}{\rho}+1}}{z^{k/\rho}} \leq r\theta \ ,$$

which yields for $p^* < \Phi_r^\rho(y) = 1/r$ that

$$\frac{\mathsf{A}_r^\rho(p,y)}{p^*} \geq \frac{1}{p^*} > \frac{1}{1/r} \geq \frac{1}{r\theta}\left(\frac{1}{ry}\right)^s > \frac{1}{r\theta}\left(\frac{p^*}{y}\right)^s = \frac{1}{r\theta}\mathcal{E}^s \ .$$

**Conclusion.** All in all, for any $y \in [1, \theta]$, it holds that

$$\frac{\mathsf{A}_r^\rho(p,y)}{p^*} \geq \frac{1}{r\theta}\mathcal{E}^s \ ,$$

with $s = \max\left(1, \frac{1}{\rho}(\frac{\ln\theta}{\ln(r\theta)} - 2)\right)$.

Finally, the threshold function $\Phi_r^\rho$ is in the class $\mathcal{P}_r$, then we have by Theorem 3.1 that $\mathsf{A}_r^\rho$ is $r$-robust, and it deduce that

$$\frac{\mathsf{A}_r^\rho(p,y)}{p^*} \geq \max\left(r, \frac{1}{r\theta}\mathcal{E}^s\right) \ ,$$

which concludes the proof.

$\square$

## B.3. Lower Bound on Smoothness

**Theorem 3.3.** *Let* $\mathsf{A}$ *be any deterministic algorithm with robustness* $r$ *and consistency* $1/r\theta$. *Suppose that* $\mathsf{A}$ *satisfies for all* $p \in [1, \theta]^n$ *and* $y \in [1, \theta]$ *that*

$$\frac{\mathsf{A}(p,y)}{p^*} \geq \max\left(r, \frac{1}{r\theta}\mathcal{E}(p^*,y)^u\right) \tag{5}$$

*for some* $u \in \mathbb{R}$, *then necessarily* $u \geq \frac{\ln\theta}{\ln(r\theta)} - 2$.

*Proof.* Let $\mathsf{A}$ be a deterministic algorithm for one-max-search with, and assume that it satisfies for all $p$ and $y$ that

$$\frac{\mathsf{A}(p,y)}{p^*} \geq \max\left(r, \frac{1}{r\theta}\mathcal{E}(y,p^*)^u\right) \ .$$

In particular, $\mathsf{A}$ has robustness $r$ and consistency $1/r\theta$.

To prove the lower bound, we will use the instances $\{\mathcal{I}_n(q)\}_{q\in[1,\theta]}$ defined in Eq. (15). On these instances, for a fixed prediction $y$, any deterministic algorithm behaves as a single threshold algorithm. Therefore, there exists $\Phi : [1,\theta] \to [1,\theta]$ satisfying that $\mathsf{A}(\mathcal{I}_n(q), y) = \mathsf{A}_\Phi(\mathcal{I}_n(q), y)$ for all $q, y \in [1,\theta]$.

The lower bound satisfied by $\mathsf{A}$ ensures that it achieves Pareto-optimal consistency $1/(r\theta)$ and robustness $r$. Consequently, $\mathsf{A}_\Phi$ also attaints them on the sequences of prices $\{\mathcal{I}_n(q)\}_{q\in[1,\theta]}$. These instances are precisely those used to establish the constraints on Pareto-optimal thresholds in Theorem 3.1, which implies that the theorem's constraints hold for $\Phi$. In particular, we have that $\Phi(r\theta) = r\theta$ and $\Phi(\theta) = 1/r$.

Let $y = \theta$ and $q = \frac{\Phi(\theta)}{1+\varepsilon} = \frac{1/r}{1+\varepsilon}$ for some $\varepsilon > 0$. Since $q < \Phi(\theta)$, when $\mathsf{A}$ is given as input the instance $\mathcal{I}_n(q)$, it does not select the maximum price and ends up selecting a price of 1, hence

$$\frac{\mathsf{A}(\mathcal{I}_n(q), \theta)}{q} = \frac{1}{q} = (1+\varepsilon)r .$$

Furthermore, by assumption, this ratio above is at least

$$\frac{1}{r\theta}\mathcal{E}(q, \theta)^u = \frac{1}{r\theta}\mathcal{E}\left(\frac{1}{(1+\varepsilon)r}, \theta\right)^u = \frac{1}{r\theta}\left(\frac{1}{(1+\varepsilon)r\theta}\right)^u = \frac{1}{(r\theta)^{u+1}}(1+\varepsilon)^u ,$$

therefore, we have that

$$(1+\varepsilon)r \geq \frac{1}{(r\theta)^{u+1}}(1+\varepsilon)^u .$$

This inequality holds for all $\varepsilon > 0$, which gives in the limit $\varepsilon \to 0$ that

$$r \geq \frac{1}{(r\theta)^{u+1}} ,$$

and we obtain by equivalences that

$$r \geq \frac{1}{(r\theta)^{u+1}} , \iff (r\theta)^u \geq \frac{\theta}{(r\theta)^2}$$
$$\iff u\ln(r\theta) \geq \ln\theta - 2\ln(r\theta)$$
$$\iff u \geq \frac{\ln\theta}{\ln(r\theta)} - 2 ,$$

which gives the claimed lower bound on $u$.

$\square$

# C. Complements to Section 4

Recall in this section the notations $P^* \sim F^*$ for the maximum price, and $Y \sim G$ for the prediction. When considered, their coupling is denoted $\pi^*$.

## C.1. Complements on Section 4.2

**Stochastic predictions, deterministic prices.**

**Corollary 4.2.** *Let $F^* = \delta_{p^*}$ for some $p^* \in [1,\theta]$, then the family $\{\mathsf{A}_r^1\}_r$ satisfies*

$$\frac{\mathbb{E}[\mathsf{A}_r^1(P^*, Y)]}{\mathbb{E}[P^*]} \geq \frac{1}{r\theta}\Lambda(p^*) . \tag{10}$$

*Proof.* This is obtained by direct instantiation of Lemma 4.1. In particular, for $F^* = \delta_{p^*}$ the second term of Eq. (8) can be substituted into with

$$\mathbb{E}[P^*\mathcal{E}(p^*, y)^s] = \mathbb{E}[P^*\mathcal{E}(P^*, Y)^s | P^* = p^*]p^* = \int \min\left\{\frac{y}{p^*}, \frac{p^*}{y}\right\}^s dG(y) .$$

Computing this integral explicitly reveals it to be $\Lambda(p^*)$. $\square$

Through this section, we will use the following identity

$$\Lambda(p^*) = \int_1^{p^*} \left(\frac{y}{p^*}\right)^s \mathrm{d}G(y) + \int_{p^*}^\theta \left(\frac{p^*}{y}\right)^s \mathrm{d}G(y).$$

(21)

which is easily derived from the proof above.

Inspection of Eq. (21) reveals that $\Lambda$ contains two different regimes (above and below $p^*$). The *mirrored* coefficients of $(p^*, y)$ in each term reflect the inherent asymmetry of a threshold algorithm: performance is highly sensitive to whether $p^* \leq \Phi_r^1(Y)$, which transfers to Eq. (21) via the definition of $\mathcal{E}$.

**Example 4.3.** *Let $F^* = \delta_{p^*}$ for some $p^* \in [1, \theta]$ and $G = \mathsf{Unif}([p^* - \epsilon, p^* + \epsilon])$. There is a constant $C > 0$, dependent only on $(s, \theta)$, such that*

$$\frac{\mathbb{E}[\mathsf{A}_r^1(P^*, Y)]}{\mathbb{E}[P^*]} \geq \frac{1}{r\theta}\left(1 - \frac{s}{2p^*}\epsilon - C\epsilon^2\right),$$

(11)

*as soon as $0 < \epsilon \leq \min\{\theta - p^*, p^* - 1\}$.*

*Proof of Example 4.3.*

1. Consider first $s > 1$. Compute $p^*\Lambda$ for this choice of $G$, which yields

$$p^*\Lambda(p^*) = \frac{p^{*1-s}}{2\epsilon}\int_{p^*-\epsilon}^{p^*} y^s \mathrm{d}y + \frac{p^{*1+s}}{2\epsilon}\int_{p^*}^{p^*+\epsilon} y^{-s}\mathrm{d}y$$

$$= \frac{p^{*1-s}}{2\epsilon}\frac{p^{*1+s} - (p^* - \epsilon)^{1+s}}{1+s} + \frac{p^{*1+s}}{2\epsilon}\frac{(p^* + \epsilon)^{1-s} - p^{*1-s}}{1-s}.$$

(22)

Continuous differentiability of $p^* \mapsto p^{*1+s}$ and $p^* \mapsto p^{*1-s}$, along with Taylor's theorem, implies the existence of $(\hat{p^*}_1, \hat{p^*}_2) \in [p^* - \epsilon, p^*] \times [p^*, p^* + \epsilon]$ such that:

$$\frac{p^{*1+s} - (p^* - \epsilon)^{1+s}}{1+s} = \epsilon p^{*s} - \frac{\epsilon^2}{2}sp^{*s-1} + \frac{\epsilon^3}{6}s(s-1)\hat{p^*}_1^{s-2},$$

$$\frac{(p^* + \epsilon)^{1-s} - p^{*1-s}}{1-s} = \epsilon p^{*-s} - \frac{\epsilon^2}{2}sp^{*-1-s} - \frac{\epsilon^3}{6}s(s+1)\hat{p^*}_2^{-(2+s)}.$$

Remark that one has the remainder bounds:

$$C_1 := \frac{s(s-1)}{6}\theta^{\min\{0, 2-s\}} \leq \frac{s(s-1)}{6}\hat{p^*}_1^{\alpha-3}$$

$$C_2 := \frac{s(1+s)}{6} \leq \frac{s(1+s)}{6\hat{p^*}_2^{2+s}}.$$

Applying these bounds and the Taylor expansions of $\Phi$ to Eq. (22), yields

$$p^*\Lambda(p^*) \geq p^* - \frac{s}{2}\epsilon - Cp^*\epsilon^2$$

(23)

with $C = (C_1\theta^{-s} + C_2)/2$. Finally, injecting Eq. (23) into Corollary 4.2 and recalling that $G = \delta_{p^*}$ implies $\Gamma = c/p^*$ yields Eq. (10).

2. Now, for $s = 1$, the computation of $\Lambda$ reduces to

$$p^*\Lambda(p^*) = \frac{1}{2\epsilon}\frac{p^{*2} - (p^* + \epsilon)^2}{2} + \frac{p^{*2}}{2\epsilon}\int_{p^*}^{p^*+\epsilon}\frac{1}{y}\mathrm{d}y$$

$$= \frac{p^*}{2} + \frac{\epsilon}{4} + \frac{p^{*2}}{2\epsilon}(\log(p^* + \epsilon) - \log(p^*)).$$

Using a Taylor expansion on $\log$, for some $t \in [0,1]$, we have

$$\frac{p^{*2}}{2\epsilon}(\log(p^* + \epsilon) - \log(p^*)) \geq \frac{p^{*2}}{2\epsilon}\left(\frac{\epsilon}{p^*} - \frac{\epsilon^2}{2}\frac{1}{p^{*2}} + \frac{\epsilon^3}{6}\frac{1}{2(p^* + \epsilon t)^3}\right)$$

$$\geq \frac{p^*}{2} + \frac{\epsilon}{4} + \frac{\epsilon^2}{24}.$$

and thus, we obtain an overall bound matching Eq. (23) up to modifying $C$. $\qquad\square$

**Corollary C.1.** *Let $F^* = \delta_{p^*}$ for some $p^* \in [1, \theta]$ and $G = \mathsf{Unif}([p^*(1 - \epsilon'), p^*(1 + \epsilon')])$. There is $C' > 0$ dependent only on $(s, \theta)$ such that*

$$\frac{\mathbb{E}[\mathsf{A}_r^1(P^*, Y)]}{\mathbb{E}[P^*]} \geq \frac{1}{r\theta}\left(1 - \frac{s}{2}\epsilon - C'\epsilon^2\right), \tag{24}$$

*as soon as $0 < \epsilon' \leq \min\{1 - p^{*-1}, \theta p^{*-1} - 1\}$.*

*Proof.* Follow the proof of Example 4.3 with $\epsilon = \epsilon'p^*$. $\qquad\square$

**Corollary C.2.** *Let $(I_k)_{k=1}^K$, $I_k := [\mu_k - \epsilon_k, \mu_k + \epsilon_k]$ for $k \in [K]$ be a collection of (w.l.o.g. disjoint) sub-intervals of $[1, \theta]$. Let $F^* = \delta_{p^*}$ and let*

$$G = \sum_{k=1}^K w_k \mathsf{Unif}(I_k)$$

*be a mixture of Uniforms of the intervals $I_k$, i.e. $w_k > 0$ for all $k \in [K]$ and $\sum_k w_k = 1$. Then, there is a constant $C'' > 0$ dependent only on $(r, \theta)$ such that*

$$\frac{\mathbb{E}[\mathsf{A}_r^1(P^*, Y)]}{\mathbb{E}[P^*]} \geq \frac{1}{r\theta\mathbb{E}[P^*]}\left(w_{k^*}\left(1 - \frac{s}{2p^*}\epsilon_k\right) + \sum_{k \neq k^*} w_k \mathcal{E}(p^*, \mu_k)^s + C'' \sum_{k \in [K]} w_k \epsilon_k^2\right).$$

*in which $k^*$ denotes the index (if it exists) such that $p^* \in I_{k^*}$.*

*Proof.* Note that $G$ has density

$$y \in [1, \theta] \mapsto \sum_{k=1}^K \frac{w_k}{2\epsilon_k}\mathbb{1}_{\{y \in [\mu_k - \epsilon_k, \mu_k + \epsilon_k]\}}$$

with respect to the Lebesgue measure. We can thus express $\Lambda$ as a function of $\Lambda_k$ its analogues for each $\mathsf{Unif}(I_k)$ as

$$\Lambda(p^*) = \sum_{k=1}^K w_k \Lambda_k(p^*). \tag{25}$$

We will now be able to proceed on each $\Lambda_k$ as in the proof of the first part of Example 4.3.

Let us remark that the case of $k^*$ permits a direct application of Example 4.3.1, as $p^* \in I_k$. This readily implies a contribution to the final bound of

$$\frac{1}{r\theta\mathbb{E}[P^*]}\left(w_{k^*}(1 - \frac{s}{2p^*}\epsilon_k - C\epsilon_k^2)\right), \tag{26}$$

which is the first term of Eq. (25).

Let us now fix $k \neq k^*$. Without loss of generality, we will order the intervals, so that $k < k^*$ implies that $x < p^*$ for every $x \in I_k$ and conversely for $k > k^*$. Notice that this implies that exactly one integral in each $\Lambda_k$, $k \neq k^*$, may be non-zero, which is to say

$$p^* \Lambda_k(p^*) = \frac{p^{*1-s}}{2\epsilon_k}\frac{(\mu_k + \epsilon_k)^{1+s} - (\mu_k - \epsilon_k)^{1+s}}{1 + s}\mathbb{1}\{k < k^*\} + \frac{p^{*1+s}}{2\epsilon_k}\frac{(\mu_k + \epsilon_k)^{1-s} - (\mu_k - \epsilon_k)^{1-s}}{1 - s}\mathbb{1}\{k > k^*\}$$

for any $k \neq k^*$. Let us take each term in turn and apply Taylor's theorem following the same methodology as the proof of Example 4.3. By adding and subtracting $\mu_k^{1+s}$ (resp. $\mu_k^{1-s}$), and using Taylor's theorem yields the existence of $(C_k^{(1)}, C_k^{(2)})_{k \neq k^*}$ such that

$$\frac{(\mu_k + \epsilon_k)^{1+s} - (\mu_k - \epsilon_k)^{1+s}}{1+s} \geq 2\epsilon_k \mu_k^s + \frac{\epsilon_k^3}{6} C_k^{(1)},$$

$$\frac{(\mu_k + \epsilon_k)^{1-s} - (\mu_k - \epsilon_k)^{1-s}}{1-s} \geq 2\epsilon_k \mu_k^{-s} + \frac{\epsilon_k^3}{6} C_k^{(2)}.$$

Combining these over $k \in [K]$ yields

$$\sum_{k \neq k^*} w_k \Lambda_k(p^*) \geq p^{*1-s} \sum_{k < k^*} w_k \mu_k^s + p^{*1+s} \sum_{k > k^*} w_k \mu_k^{-s} + \frac{1}{12} \left( \sum_{k < k^*} C_k^{(1)} \theta^{1-s} w_k \epsilon_k^2 + \sum_{k > k^*} C_k^{(2)} w_k \epsilon_k^2 \right). \tag{27}$$

Now, add together Equations (25) and (26), recalling Eq. (27), and appeal to Corollary 4.2 to obtain

$$\frac{\mathbb{E}[\mathsf{A}_r^1(P^*, Y)]}{\mathbb{E}[P^*]} \geq \Gamma \left( w_{k^*} \left( 1 - \frac{s}{2p^*} \epsilon_k \right) + \sum_{k < k^*} w_k \left( \frac{\mu_k}{p^*} \right)^s + \sum_{k > k^*} w_k \left( \frac{p^*}{\mu_k} \right)^s + C'' \sum_{k \in [K]} \epsilon_k^2 \right) \tag{28}$$

for a suitably chosen constant $C'' \geq 0$. To complete the proof, notice that the two sums can be combined using $\mathcal{E}$ as $\mu_k < p^*$ if an only if $k < k^*$. $\qquad\square$

### Deterministic predictions, stochastic prices

Hereafter, we will use the following identity

$$\Upsilon(y) := \frac{1}{\mathbb{E}[P^*]} p^* \int_1^y \left( \frac{p^*}{y} \right)^s \mathrm{d}F^*(p^*) + p^* \int_y^\theta \left( \frac{y}{p^*} \right)^s \mathrm{d}F^*(p).$$

### Stochastic independent predictions and prices

**Corollary 4.5.** *Let $\pi^* = F^* \otimes G$, the family $\{\mathsf{A}_r^1\}_r$ satisfies*

$$\frac{\mathbb{E}[\mathsf{A}_r^1(P^*, Y)]}{\mathbb{E}[P^*]} \geq \frac{1}{r\theta} \int \Upsilon(y) \mathrm{d}G(y)$$

$$= \frac{1}{r\theta} \int \frac{p^* \Lambda(p^*)}{\mathbb{E}[P^*]} \mathrm{d}F^*(p^*). \tag{13}$$

*Proof.* Starting from Lemma 4.1, this follows from

$$\frac{\mathbb{E}[\mathsf{A}_r^1(P^*, Y)]}{\mathbb{E}[P^*]} \geq \frac{1}{r\theta\mathbb{E}[P^*]} \int \int p^* \mathcal{E}(p^*, y)^s \mathrm{d}G(y) \mathrm{d}F^*(p^*).$$

$\qquad\square$

**Corollary C.3.** *Let $\pi^* = F^* \otimes G$, with $F^*, G$ having density with respect to the Lebesgue measure, then the family $\{\mathsf{A}_r^1\}_r$ satisfies*

$$\frac{\mathbb{E}[\mathsf{A}_r^1(P^*, Y)]}{\mathbb{E}[P^*]} \geq \frac{1}{r\theta\mathbb{E}[P^*]} \left( \int_1^\theta p^{*1-s} \mathrm{d}F^*(p^*) \int_1^\theta y^{-s} \mathrm{d}G(y) + \int_1^\theta p^{*1+s} \mathrm{d}F^*(p^*) \int_1^\theta y^s \mathrm{d}G(y) \right.$$

$$\left. - \int_1^\theta \left( y^s \int_1^y p^{*1-s} \mathrm{d}F^*(p^*) + y^{-s} \int_y^\theta p^{*1+s} \mathrm{d}F^*(p^*) \right) \mathrm{d}G(y) \right) \tag{29}$$

*Remark* C.4. The result of Corollary C.3 can be slightly tweaked to hold even without densities using Lebesgue-Stieltjes integration by parts. We omit these details for the sake of conciseness.

*Proof.* Starting with Corollary 4.5, decompose the integral as

$$\int_1^\theta p^* \Lambda(p^*) \mathrm{d}F^*(p^*) = \underbrace{\int_1^\theta p^{*1-s} \int_1^{p^*} y^s \mathrm{d}G(y) \mathrm{d}F^*(p^*)}_{A} + \underbrace{\int_1^\theta p^{*1+s} \int_{p^*}^\theta y^{-s} \mathrm{d}G(y) \mathrm{d}F^*(p^*)}_{B} .$$

Using integration by parts, in which the parts for A are $x \mapsto \int_1^x p^{*1-s} \mathrm{d}F^*(p^*)$ and $x \mapsto \int_1^x y^s \mathrm{d}G(y)$ yields

$$A = \int_1^\theta p^{*1-s} \mathrm{d}F^*(p^*) \int_1^\theta y^{-s} \mathrm{d}G(y) - \int_1^\theta y^s \int_1^y p^{*1-s} \mathrm{d}F^*(p^*) \mathrm{d}G(y) . \tag{30}$$

Similarly, $B$ can be integrated by parts with parts $x \mapsto -\int_x^\theta p^{*1+s} \mathrm{d}F^*(p^*)$ and $x \mapsto \int_x^\theta y^{-s} \mathrm{d}G(y)$, which yields

$$B = \int_1^\theta p^{*1+s} \mathrm{d}F^*(p^*) \int_1^\theta y^s \mathrm{d}G(y) - \int_1^\theta y^{-s} \int_y^\theta p^{*1+s} \mathrm{d}F^*(p^*) \mathrm{d}G(y) . \tag{31}$$

Combining Equations (30) with (31) completes the proof. □

**Proposition C.5.** *Let $\pi^* = F^* \otimes G$ and $F^* = G = \mathsf{Unif}([c_1, c_2])$ the family $\{\mathsf{A}_r^1\}_r$ satisfies*

$$\frac{\mathbb{E}[\mathsf{A}_r^1(P^*, Y)]}{\mathbb{E}[P^*]} \geq \frac{1}{r\theta} \frac{2}{\zeta(1)^3} \left( \zeta(2-s)\zeta(1-s) - \frac{c_1^{2-s}\zeta(1+s)}{2-s} \right.$$

$$+ \zeta(2+s)\zeta(1+s) - \frac{c_2^{2+s}\zeta(1-s)}{2+s}$$

$$\left. - \zeta(3)\left( \frac{1}{2-s} - \frac{1}{2+s} \right) \right)$$

*when $s \notin \{1, 2\}$, with $\zeta : \gamma \in (0, +\infty) \mapsto (c_2^\gamma - c_1^\gamma)\gamma^{-1}$.*

*Proof.* Starting with the decomposition of Corollary C.3, we can compute the terms separately. For the first two, we have

$$\int_1^\theta p^{*1-s} \mathrm{d}F^*(p^*) \int_1^\theta y^{-s} \mathrm{d}G(y) = C \left( \frac{c_2^{2-s} - c_1^{2-s}}{2-s} \right) \left( \frac{m_2^{1-s} - m_1^{1-s}}{1-s} \right) \tag{32}$$

$$\int_1^\theta p^{*\alpha} \mathrm{d}F^*(p^*) \int_1^\theta y^s \mathrm{d}G(y) = C \left( \frac{c_2^{2+s} - c_1^{2+s}}{2+s} \right) \left( \frac{m_2^{1+s} - m_1^{1+s}}{1+s} \right) \tag{33}$$

in which

$$C := \frac{1}{(c_2 - c_1)(m_2 - m_1)} .$$

Turning now to the second term of Eq. (29), we have

$$\int_1^y p^{*1-s} \mathrm{d}F^*(p^*) = \frac{1}{c_2 - c_1} \left( \frac{y^{2-s} - c_1^{2-s}}{2-s} \mathbb{1}_{\{y \in [c_1, c_2]\}} + \frac{c_2^{2-s} - c_1^{2-s}}{2-s} \mathbb{1}_{\{y > c_2\}} \right)$$

and

$$\int_y^\theta p^{*\alpha} \mathrm{d}F^*(p^*) = \frac{1}{c_2 - c_1} \left( \frac{c_2^{2+s} - y^{2+s}}{2+s} \mathbb{1}_{\{y \in [c_1, c_2]\}} + \frac{c_2^{2+s} - c_1^{2+s}}{2+s} \mathbb{1}_{\{y < c_1\}} \right) ,$$

so that, by integrating according to Eq. (29) yields

$$\int_1^\theta y^s \int_1^y p^{*1-s} \mathrm{d}F^*(p^*) \mathrm{d}G(y) = C \left( \frac{1}{2-s} \left[ \frac{y^3}{3} - c_1^{2-s} \frac{y^{1+s}}{1+s} \right]_{c_1 \vee m_1 \wedge c_2}^{c_2 \wedge m_2 \vee c_1} + \frac{c_2^{2-s} - c_1^{2-s}}{2-s} \left[ \frac{y^{1+s}}{\alpha} \right]_{c_2 \vee m_1}^{c_2 \vee m_2} \right) \tag{34}$$

$$\int_1^\theta y^{-s} \int_y^\theta p^{*1+s} \mathrm{d}F^*(p^*) \mathrm{d}G(y) = C \left( \frac{1}{2+s} \left[ c_2^{2+s} \frac{y^{1-s}}{1-s} - \frac{y^3}{3} \right]_{c_1 \vee m_1 \wedge c_2}^{c_2 \wedge m_2 \vee c_1} + \frac{c_2^{2+s} - c_1^{2+s}}{2+s} \left[ \frac{y^{1-s}}{1-s} \right]_{c_1 \wedge m_1}^{c_1 \wedge m_2} \right) \tag{35}$$

Recombining Equations (32)–(35) yields the result, up to simplifying for $m_i = c_i$ $i \in \{1, 2\}$. □

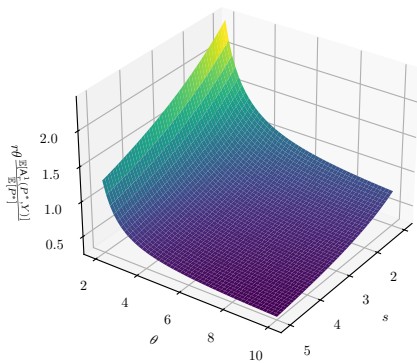

*Figure 5.* Numerical quadrature of Proposition C.5 for $c_1 = 1$, $c_2 = \theta$ (for $\theta \in [1, 10]$) as a function of $s \in [1, 5]$.

### C.2. Complements to section 4.3

Since it holds regardless of the coupling $\pi^*$, the bound Eq. (14) has two direct benefits. First, it provides a notion of robustness for uncertainty in the coupling which is relevant for risk-assessment in practical applications. Second, it isolates the influence of the marginal distributions on the prediction from the coupling of $Y$ and $P^*$. Consequently, we can return to (8) and isolate the contribution of the coupling, either through its transport sub-optimality

$$\mathbb{E}[P^*\mathcal{E}(Y, P^*)^{\alpha-1}] - \inf_{\pi \in \Pi(G, F^*)} \mathbb{E}[P^*\mathcal{E}(Y, P^*)^{\alpha-1}]$$

or through a multiplicative analogue

$$\frac{\mathbb{E}[P^*\mathcal{E}(Y, P^*)^{\alpha-1}]}{\inf_{\pi \in \Pi(G, F^*)} \mathbb{E}[P^*\mathcal{E}(Y, P^*)^{\alpha-1}]}.$$

The geometry of these objects is highly intricate and unfortunately doesn't appear to have been studied previously. This highlights an interesting direction of research in the competitive analysis of optimal transport.

**Duality**. The Kantorovich problem Eq. (14) admits a dual problem under general conditions (see e.g. Thm. 5.10 in (Villani, 2009)). In our case, the cost function is $c := (p^*, y) \in [1, \theta]^2 \mapsto p^*\mathcal{E}(p^*, y)^s \in [0, \theta]$. This duality is strong, meaning that

$$\inf_{\pi \in \Pi(F^*, G)} \int p^*\mathcal{E}(p^*, y)^s \mathrm{d}\pi(p^*, y) = \sup_{(\varphi, \psi) \in \Xi} \int \varphi(G)\mathrm{d}G(y) + \int \psi(p^*)\mathrm{d}F^*(y), \tag{36}$$

in which

$$\Xi := \{(\varphi, \psi) : [1, \theta]^2 \to [0, +\infty)^2 \text{ bounded and measurable} : \forall (p^*, y) \in [1, \theta]^2 \quad \varphi(p^*) + \psi(y) \le c(p^*, y)\}.$$

Given a bounded measurable function $f$, let $f^c$ denote its $c$-transform, i.e. the operator $\cdot^c$ such that maps $f$ to

$$f^c : p^* \mapsto \inf_{y \in [1, \theta]} p^*\mathcal{E}(p^*, y)^s - \varphi(y). \tag{37}$$

The definition Eq. (37) shows that any bounded measurable function $\varphi : [1, \theta] \to \mathbb{R}$ forms an admissible $(\varphi, \varphi^c) \in \Xi$ with its $c$-transform (this is symmetrical in the sense that $(\psi^c, \psi)$ is admissible if $\psi$ is bounded and measurable).

From the perspective of competitive analysis, the dual problem offers an appealing tool, as it suffices to propose a potential $\varphi$, compute its $c$-transform, and integrate it to obtain a bound. Of course, guessing the optimal potential $\varphi$ is as hard as solving the primal, but sub-optimal proposals can effectively leverage insights about the problem. Proposition 4.7 gives an example of this methodology. Note that it is possible to improve the potential $\varphi$ again by proposing $(\varphi^c)^c$, which we omit for brevity.

**Proposition 4.7.** *The family* $\{\mathsf{A}_r^1\}_r$ *satisfies*

$$\frac{\mathbb{E}[\mathsf{A}_r^1(P^*, Y)]}{\mathbb{E}[P^*]} \geq \frac{1}{r\theta} \frac{\int_1^{\theta^{\frac{1}{2}}} p^{*1+s} \mathrm{d}F^*(p^*) + \int_{\theta^{\frac{1}{2}}}^{\theta} p^{*1-s} \mathrm{d}F^*(p^*)}{\mathbb{E}[P^*]} \, .$$

*Moreover, the RHS is the infimum over $G$ of Eq.* (14).

*Proof.* We split the proof into two parts, starting with the bound, and then the equality.

1. We start with the potential $\psi \equiv 0$, whose $c$-transform (see Eq. (37)) is

$$\varphi(p^*) = \psi^c(p^*) = \inf_{y \in [1,\theta]} p^* \mathcal{E}(p^*, y)^s$$

$$= \min \left\{ \frac{1}{p^*}, \frac{p^*}{\theta} \right\}^s p^*$$

$$= p^{*1-s} \mathbb{1}_{\{p^* \geq \sqrt{\theta}\}} + p^{*1+s} \mathbb{1}_{\{p^* \leq \sqrt{\theta}\}} \, .$$

   Inserting into Eq. (36) yields the result.

2. By duality of the optimal transport problem,

$$\inf_{G \in \mathscr{P}([1,\theta])} \inf_{\pi \in \Pi(F^*, G)} \int c(p^*, y) \mathrm{d}\pi(p^*, y) = \inf_{G \in \mathscr{P}([1,\theta])} \sup_{(\varphi, \psi) \in \Xi} \int \varphi(p^*) \mathrm{d}F^*(p^*) + \int \psi(y) \mathrm{d}G(y)$$

$$\geq \sup_{(\varphi, \psi) \in \Xi} \left\{ \int \varphi(p^*) \mathrm{d}F^*(p^*) + \inf_{G \in \mathscr{P}([1,\theta])} \int \psi(y) \mathrm{d}G(y) \right\}. \qquad (38)$$

   The inner infimum in Eq. (38) is equal to $\iota := \inf\{\psi(y) : y \in [1,\theta]\} \in \mathbb{R}$. Consequently, the constraint set $\Xi$ can be replaced[3] without changing the value of the problem by

$$S := \{(\varphi, \iota) \in [0, +\infty)^{[1,\theta]} \times \mathbb{R} : \quad \forall p^* \in [1,\theta]^2 \quad \varphi(p^*) \leq \inf_{y \in [1,\theta]} c(p^*, y) - \iota\}$$

   so that

$$\sup_{(\varphi, \psi) \in \Xi} \left\{ \int \varphi(p^*) \mathrm{d}F^*(p^*) + \inf_{G \in \mathscr{P}([1,\theta])} \int \psi(y) \mathrm{d}G(y) \right\} = \sup_{(\varphi, \iota) \in S} \left\{ \int \varphi(p^*) \mathrm{d}F^*(p^*) + \iota \right\}$$

$$= \sup_{\iota \in \mathbb{R}} \sup_{\psi_\iota \in S_\iota} \int \varphi_\iota(p^*) \mathrm{d}F^*(p^*) + \iota \, , \qquad (39)$$

   in which $S_\iota := \{\psi_\iota : \psi_\iota \leq \inf_{y \in [1,\theta]} c(p^*, y) - \iota\}$.

   As $F^*$ is positive, the inner maximisation over $\psi_\iota$ in Eq. (39) saturates the constraints, whereafter, since $\int \iota \mathrm{d}F^* = \iota$ and by combining with Eq. (38), one has the result. $\qquad \square$

# D. Additive Prediction Error

**Theorem 3.4.** *Let* $\mathsf{A}$ *be any deterministic algorithm with robustness $r$ and consistency $1/r\theta$. Suppose that $\mathsf{A}$ satisfies for all* $p \in [1,\theta]^n$ *and* $y \in [1,\theta]$ *that*

$$\frac{\mathsf{A}(p, y)}{p^*} \geq \max\left(r, \frac{1}{r\theta} - \beta \frac{\eta(p^*, y)}{p^*}\right) \qquad (6)$$

*for some $\beta \geq 0$, then necessarily $\beta \geq \beta^*$, where*

$$\beta^* := \frac{1 - r^2\theta}{r\theta} \max\left(\frac{1}{1-r}, \frac{1}{r\theta - 1}\right).$$

*Moreover, Algorithm* $\mathsf{A}_r^1$ *satisfies* (6) *with $\beta = \beta^*$, which shows its optimality.*

---

[3] Hereafter, all optimisation problems are over functions $(\varphi, \psi)$ which are bounded and measurable. We omit this line by line to reduce notational clutter.

We separately prove the lower and upper bounds stated in the theorem. First, in Lemma D.2, we establish the lower bound on the performance of $\mathsf{A}_r^1$:

$$\frac{\mathsf{A}_r^1(p,y)}{p^*} \geq \max\left(r, \frac{1}{r\theta} - \beta^* \frac{\eta(p^*,y)}{p^*}\right) .$$

Next, in Lemma D.3, we show that $\beta^*$ is the best possible constant.

### D.1. Smoothness guarantee on $\mathsf{A}_r^1$

We begin by proving a lemma that will be useful for establishing the smoothness of $\mathsf{A}_r^1$.

**Lemma D.1.** *The function* $\varphi_r : z \mapsto \frac{r\theta-1}{1-r} + \frac{1-r^2\theta}{1-r} \cdot \frac{z}{r\theta}$ *satisfies for all $z$ that*

$$\varphi(z) - r\theta = \frac{1-r^2\theta}{r\theta-1}(z - \varphi_r(z)) .$$

*Proof.* This lemma can be proved with immediate computation. For all $z \in \mathbb{R}$, it holds that

$$
\begin{aligned}
z - \varphi_r(z) &= z - \frac{r\theta-1}{1-r} + \frac{1-r^2\theta}{1-r} \cdot \frac{z}{r\theta} \\
&= \frac{r\theta - r^2\theta = 1 + r^2\theta}{1-r} \cdot \frac{z}{r\theta} - \frac{r\theta-1}{1-r} \\
&= \frac{r\theta-1}{1-r}\left(\frac{z}{r\theta} - 1\right) .
\end{aligned}
$$

On the other hand, we have that

$$
\begin{aligned}
\varphi_r(z) &= \frac{r\theta-1}{1-r} + \frac{1-r^2\theta}{1-r} \cdot \frac{z}{r\theta} - r\theta \\
&= \frac{r\theta-1-r\theta+r^2\theta}{1-r} + \frac{1-r^2\theta}{1-r} \cdot \frac{z}{r\theta} \\
&= \frac{1-r^2\theta}{1-r}\left(\frac{z}{r\theta} - 1\right) \\
&= \frac{1-r^2\theta}{r\theta-1} \cdot \frac{r\theta-1}{1-r}\left(\frac{z}{r\theta} - 1\right) \\
&= \frac{1-r^2\theta}{r\theta-1}(z - \varphi_r(z)) ,
\end{aligned}
$$

which concludes the proof. $\square$

**Lemma D.2.** *Algorithm $\mathsf{A}_r^1$ satisfies*

$$\frac{\mathsf{A}_r^1(p,y)}{p^*} \geq \max\left(r, \frac{1}{r\theta} - \beta\frac{\eta(p^*,y)}{p^*}\right) , \tag{40}$$

*where* $\beta := \frac{1-r^2\theta}{r\theta} \max\left(\frac{1}{1-r}, \frac{1}{r\theta-1}\right) .$

*Proof.* By Theorem 3.1 and by definition of $\mathsf{A}_r^1$, we have that $\mathsf{A}_r^1$ is $r$-robust, hence it satisfies for all $p \in [1,\theta]^n$ and $y \in [1,\theta]$ that $\frac{\mathsf{A}_r^1}{p^*} \geq r$. It remains to prove the second lower bound that characterizes smoothness. We will prove it separately for $y \in [1, r\theta]$ and $y \in [r\theta, \theta]$.

**Case 1.** For $y \in [1, r\theta]$, the acceptance threshold is $\Phi_r^1(y) = r\theta$. If $p^* < r\theta$ then

$$\frac{\mathsf{A}_r^1(p,r)}{p^*} \geq \frac{1}{p^*} \geq \frac{1}{r\theta} \geq \frac{1}{r\theta} - \beta\frac{\eta(p^*,y)}{p^*} .$$

Assume now that $p^* > r\theta$. It holds that

$$\frac{\mathsf{A}_r^1(p,r)}{p^*} \geq \frac{r\theta}{p^*} . \tag{41}$$

Since $r\theta \geq 1$, we have that $(r\theta)^2 \geq r\theta$, thus the mapping $z \mapsto \frac{z - r^2\theta^2}{z - r\theta}$ is non-decreasing on $(r\theta, \theta]$, and we deduce that

$$\beta = \frac{1 - r^2\theta}{r\theta} \max\left(\frac{1}{1-r}, \frac{1}{r\theta - 1}\right) \geq \frac{1}{r\theta} \cdot \frac{1 - r^2\theta}{1 - r} = \frac{1}{r\theta} \cdot \frac{\theta - r^2\theta^2}{\theta - r\theta} \geq \frac{1}{r\theta} \cdot \frac{p^* - r^2\theta^2}{p^* - r\theta} ,$$

and successive equivalences, recalling that $p^* > r\theta$, show that

$$\beta \geq \frac{1}{r\theta} \cdot \frac{p^* - r^2\theta^2}{p^* - r\theta} \iff \beta(p^* - r\theta) \geq \frac{1}{r\theta}(p^* - r^2\theta^2)$$

$$\iff \beta\left(1 - \frac{r\theta}{p^*}\right) \geq \frac{1}{r\theta}\left(1 - \frac{r^2\theta^2}{p^*}\right)$$

$$\iff \frac{r\theta}{p^*} \geq \frac{1}{r\theta} - \beta\left(1 - \frac{r\theta}{p^*}\right) .$$

Combining this with Eq. (41) and using that $y \leq r\theta$, we obtain

$$\frac{\mathsf{A}_r^1(p,r)}{p^*} \geq \frac{1}{r\theta} - \beta\left(1 - \frac{r\theta}{p^*}\right) \geq \frac{1}{r\theta} - \beta\left(1 - \frac{y}{p^*}\right) = \frac{1}{r\theta} - \beta\frac{\eta(p^*,y)}{p^*} .$$

**Case 2.**  For $y \in (r\theta, \theta]$, the threshold is $\Phi_r^1(y) = \frac{r\theta - 1}{1 - r} + \frac{1 - r^2\theta}{1 - r} \cdot \frac{y}{r\theta}$.

Assume that $p^* < \Phi_r^1(y)$. Since $y > r\theta$, using the definition of $\beta$ and Lemma D.1 yields

$$\beta = \frac{1 - r^2\theta}{r\theta} \max\left(\frac{1}{1-r}, \frac{1}{r\theta - 1}\right) \geq \frac{1}{r\theta} \cdot \frac{1 - r^2\theta}{r\theta - 1} = \frac{1}{r\theta} \cdot \frac{\varphi_r(y) - r\theta}{y - \varphi_r(y)} .$$

Using again that $y > r\theta$, we have that the mapping $z \mapsto \frac{z - r\theta}{y - z}$ is non-decreasing on $[1, y]$, and given that $y < \Phi_r^1(y) = \varphi_r(y)$ and both $y$ and $\varphi_r(y)$ are within the interval $[1, y]$ ($\varphi(z) \leq z$ for $z \geq r\theta$), we deduce that

$$\beta \geq \frac{1}{r\theta} \cdot \frac{p^* - r\theta}{y - p^*} ,$$

and using that $y < \Phi_r^1(y) \leq y$, this is equivalent to writing

$$\beta\frac{y - p^*}{p^*} \geq \frac{1}{r\theta} - \frac{1}{p^*} ,$$

Hence we have the lower-bound

$$\frac{\mathsf{A}_r^1(p,y)}{p^*} \geq \frac{1}{p^*} \geq \frac{1}{r\theta} - \beta\frac{y - p^*}{p^*} = \frac{1}{r\theta} - \beta\frac{\eta(y, p^*)}{p^*} .$$

Assume now that $p^* \in [\Phi_r^1(y), y)$. Since $\mathsf{A}_r^1$ is $r$-robust and $1/r\theta$-consistent, Theorem 3.1 shows that the threshold $\Phi_r^1$ satisfies $\Phi_r^1(z) \geq \frac{z}{r\theta}$ for all $z \in (r\theta, \theta]$, which is in particular true for $y$ with the current assumptions. Therefore, we obtain immediately that

$$\frac{\mathsf{A}_r^1(p,y)}{p^*} \geq \frac{\Phi_r^1}{p^*} \geq \frac{1}{r\theta} \geq \frac{1}{r\theta} - \beta\frac{\eta(p^*,y)}{p^*} .$$

Finally, assume that $p^* \in [y, \theta]$. Using the expression of $\Phi_r^1(y)$, we have

$$
\begin{aligned}
\frac{A_r^1(p, y)}{p^*} &\geq \frac{\Phi_r^1}{p^*} = \frac{1 - r\theta}{1 - r} \cdot \frac{1}{p^*} + \frac{1 - r^2\theta}{(1 - r)r\theta} \cdot \frac{y}{p^*} \\
&\geq \frac{1 - r\theta}{1 - r} \cdot \frac{1}{\theta} + \frac{1 - r^2\theta}{(1 - r)r\theta} \cdot \frac{y}{p^*} \\
&= \frac{1 - r\theta}{1 - r} \cdot \frac{1}{\theta} + \frac{1 - r^2\theta}{(1 - r)r\theta} - \frac{1 - r^2\theta}{(1 - r)r\theta} \left( 1 - \frac{y}{p^*} \right) \\
&= \frac{1}{r\theta} \cdot \frac{r^2 - r + 1 - r^2\theta}{1 - r} - \frac{1 - r^2\theta}{(1 - r)r\theta} \left( 1 - \frac{y}{p^*} \right) \\
&= \frac{1}{r\theta} - \frac{1 - r^2\theta}{(1 - r)r\theta} \left( 1 - \frac{y}{p^*} \right) \\
&\geq \frac{1}{r\theta} - \beta \frac{\eta(p^*, y)}{p^*} ,
\end{aligned}
$$

where we used in the last inequality that $\beta := \frac{1 - r^2\theta}{r\theta} \max \left( \frac{1}{1-r}, \frac{1}{r\theta - 1} \right) \geq \frac{1 - r^2\theta}{(1-r)r\theta}$ and that $\eta(p^*, y) = p^* - y$ since $p^* \geq y$. This concludes the proof. □

### D.2. Lower bound on smoothness

**Lemma D.3.** *Let* A *be any algorithm with robustness $r$ and consistency $1/r\theta$. Suppose that* A *satisfies for all $p \in [1, \theta]^n$ and $y \in [1, \theta]$ that*

$$
\frac{A(p, y)}{p^*} \geq \max \left( r, \frac{1}{r\theta} - \beta \frac{\eta(p^*, y)}{p^*} \right) , \tag{42}
$$

*for some $\beta \in \mathbb{R}$, then necessarily $\beta \geq \frac{1 - r^2\theta}{r\theta} \max \left( \frac{1}{1-r}, \frac{1}{r\theta - 1} \right)$.*

*Proof.* Consider an algorithm A and $\beta \in \mathbb{R}$ satisfying the assumptions of the theorem. To establish the lower bound, we consider the instances $\{\mathcal{I}_n(q)\}_{q \in [1, \theta]}$ as defined in Eq. (15). On these instances, any deterministic algorithm is equivalent to a threshold-based algorithm. In particular, A is identical to $A_\Phi$ for some $\Phi : [1, \theta] \to [1, \theta]$.

The assumption on A ensures that it achieves Pareto-optimal consistency $1/(r\theta)$ and robustness $r$. Consequently, $A_\Phi$ also attains them on the sequences of prices $\{\mathcal{I}_n(q)\}_{q \in [1, \theta]}$. These instances are precisely those used to establish the constraints on Pareto-optimal thresholds in Theorem 3.1, which implies that the theorem's constraints hold for $\Phi$. In particular, we have that $\Phi(r\theta) = r\theta$ and $\Phi(\theta) = 1/r$.

Let us now prove the lower bound on $\beta$. Let $y = \theta$ and $q_\varepsilon = \frac{\Phi(\theta)}{1 + \varepsilon} < \Phi(\theta)$ for some $\varepsilon > 0$. Using Eq. (16) and the assumed lower bound on A, it holds that

$$
\frac{A(\mathcal{I}_n(q_\varepsilon), y)}{p^*} = \frac{A_\Phi(\mathcal{I}_n(q_\varepsilon), y)}{q_\varepsilon} = \frac{1}{q_\varepsilon} \geq \frac{1}{r\theta} - \beta \cdot \frac{\theta - q_\varepsilon}{q_\varepsilon} .
$$

Taking the limit when $\varepsilon \to 0$ and recalling that $\Phi(\theta) = 1/r$, we obtain that

$$
r \geq \frac{1}{r\theta} - \beta \cdot \frac{\theta - 1/r}{1/r} = \frac{1}{r\theta} - \beta \cdot (r\theta - 1) ,
$$

and it follows that

$$
\beta \geq \frac{1/r\theta - r}{r\theta - 1} = \frac{1 - r^2\theta}{r\theta(r\theta - 1)} . \tag{43}
$$

On the other hand, for $y = r\theta$ and $q = \theta$, using Eq. (16), the assumed lower bound on A, and that $\Phi(r\theta) = r\theta$, we have

$$
\frac{A_\Phi(\mathcal{I}_n(\theta), y)}{\theta} = \frac{A_\Phi(\mathcal{I}_n(\theta), y)}{\theta} = \frac{\Phi(r\theta) + O(\frac{1}{n})}{\theta} = r + O(\tfrac{1}{n}) \geq \frac{1}{r\theta} - \beta \cdot \frac{\theta - r\theta}{\theta} = \frac{1}{r\theta} - \beta(1 - r) .
$$

Taking the limit for $n \to \infty$, we obtain that

$$\beta \geq \frac{1/r\theta - r}{1 - r} = \frac{1 - r^2\theta}{r\theta(1 - r)}. \tag{44}$$

Finally, combining Equations (43) and (44), we deduce that

$$\beta \geq \max\left(\frac{1 - r^2\theta}{r\theta(r\theta - 1)}, \frac{1 - r^2\theta}{r\theta(1 - r)}\right) = \frac{1 - r^2\theta}{r\theta} \max\left(\frac{1}{r\theta - 1}, \frac{1}{1 - r}\right).$$

$\square$

### D.3. Comparison with prior smooth algorithms

In (Benomar & Perchet, 2025), the authors introduce a randomized family $\{\tilde{A}^\rho\}_{\rho \in [0,1]}$ of algorithms. For a fixed $\rho$, the maximum robustness that their algorithm can achieve is at most $\frac{1 - e^\rho}{\rho}\theta^{-1/2}$, hence remains bounded away from $\theta^{-1/2}$. Given any robustness level $r \in [\theta^{-1}, \frac{1 - e^\rho}{\rho}\theta^{-1/2}]$, the corresponding consistency achieved by their algorithm is $c = \left(\frac{1 - e^\rho}{\rho}\right)^2 \frac{1}{r\theta}$. This algorithm ensures smoothness in expectation with respect to the error $\eta(p^*, y)$, i.e. that

$$\frac{\mathbb{E}[\tilde{A}^\rho(p, y)]}{p^*} \geq \left(\frac{1 - e^\rho}{\rho}\right)^2 \frac{1}{r\theta} - \beta_\rho \frac{\eta(p^*, y)}{p^*},$$

with $\beta_\rho$ a constant proportional to $1/\rho$. The major drawbacks of this approach are that

1. the achieved robustness and consistency are not Pareto-optimal, i.e. they deviate from the front defined in Eq. (1),

2. the guarantees of the algorithm only hold in expectation, since the algorithm is randomized.

### D.4. Probabilistic analysis

**Corollary D.4.** *The family $\{A_r^1\}_r$ satisfies*

$$\frac{\mathbb{E}[A_r^1(P^*, Y)]}{\mathbb{E}[P^*]} \geq \max\left\{r, \frac{1}{r\theta} - \beta\frac{\mathbb{E}[P^*\eta(P^*, Y)]}{\mathbb{E}[P^*]}\right\}. \tag{45}$$

**Corollary D.5.** *The family $\{A_r^1\}_r$ satisfies the worst-case performance ratio bound*

$$\frac{\mathbb{E}[A_r^1(P^*, Y)]}{\mathbb{E}[P^*]} \geq \frac{1}{r\theta} - \beta \sup_{\pi \in \Pi(F^*, G)} \frac{\int p^*\eta(p^*, y)\mathrm{d}\pi(p^*, y)}{\mathbb{E}[P^*]}. \tag{46}$$

One can observe that the bounds of Corollary D.5 represent the supremum version of the transport problem associated with $\mathscr{W}_1$ (see below). The supremum is expected due to the additive nature of the analysis, which makes the error term a negative (additive) correction rather than a multiplicative factor. While not a classical optimal transport problem, this supremum can be transformed into an optimal transport problem with cost $(y, p^*) \mapsto -|y - p^*|$ and one can recover (parts of) the standard theory from there, see e.g. (Villani, 2009).

Note that the Wasserstein-$p$ distance, for $p \in [1, +\infty)$, denoted $\mathscr{W}_p$, on the space $\mathscr{P}([1, \theta])$ of probability distributions[4] over $[1, \theta]$ is

$$\mathscr{W}_p : (F^*, G) \mapsto \inf_{\pi \in \Pi(F^*, G)} \int |p^* - y|^p \, \mathrm{d}\pi(p^*, y).$$

# E. Additional Numerical Experiments

We give here an additional experiment made with the same synthetic data described in Section 5. While the first experiment shows the performance of the algorithm as a function of the multiplicative error. Instead of fixing $\mathcal{E}_{\min}$, we set a maximum

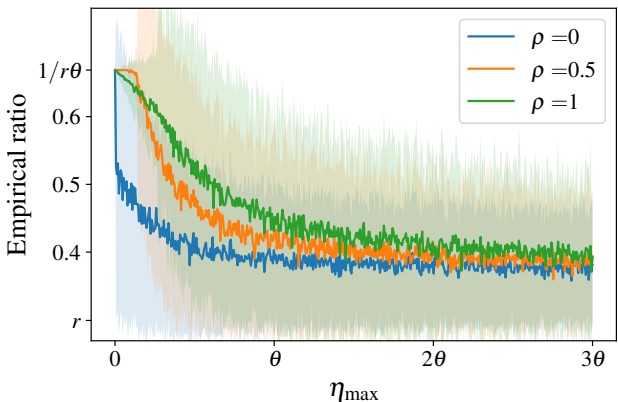

*Figure 6.* Performance of $\mathsf{A}_r^\rho$ with $\rho \in \{0, 0.5, 1\}$.

error level $\eta_{\max}$ and sample $y$ uniformly at random from the interval $[p^* - \eta_{\max}, p^* + \eta_{\max}]$. Figure 6 presents the results in this setting.

Similarly to the behaviour with respect to the multiplicative error, Figure 6 shows that $\rho = 0$ yields a significant performance degradation for an arbitrarily small error, which confirms the brittleness of Sun et al. (2021)'s algorithm. In contrast, $\rho = 1$ achieves the best smoothness, having a performance that gracefully degrades with the prediction error.

### E.1. Experiments on real datasets

We use the same experimental setting and Bitcoin data as in Section 5, but we set different values of $\lambda \in \{0.2, 0.8\}$ instead of fixing $\lambda = 0.5$ as in Figure 4. This yields different robustness levels, again expressed as $r = \theta^{-(1-\lambda/2)}$. The results are shown in Figures 7 and 8 for $\lambda = 0.2$ and $0.8$, respectively.

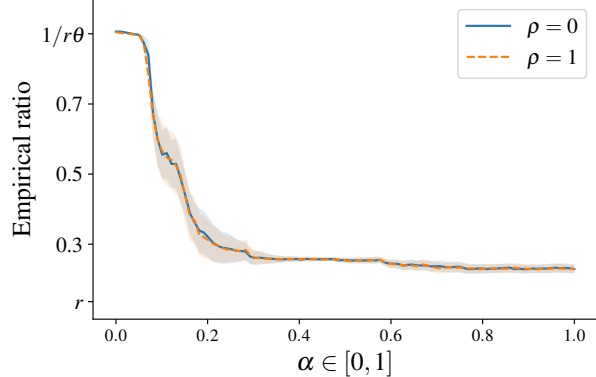

*Figure 7.* Comparison of $\mathsf{A}_r^1$ and $\mathsf{A}_r^0$ on the Bitcoin price dataset with $\lambda = 0.2$.

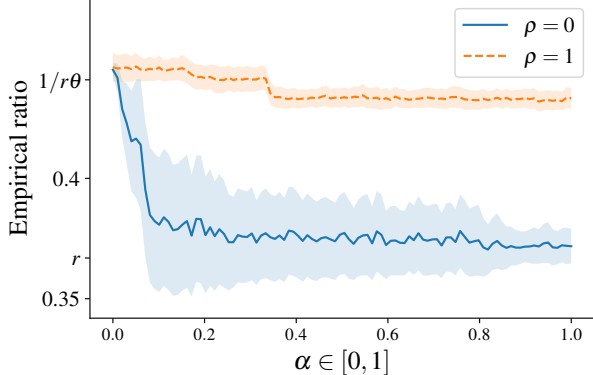

*Figure 8.* Comparison of $\mathsf{A}_r^1$ and $\mathsf{A}_r^0$ on the Bitcoin price dataset with $\lambda = 0.8$.

For $\lambda = 0.2$, Figure 7 shows that the performances of $\mathsf{A}_r^1$ and $\mathsf{A}_r^0$ are similar when $\lambda$ is small, i.e., when $r$ is close to $1/\theta$. This corresponds to a consistency of $1$, meaning that the algorithm fully trusts the prediction. Since both algorithms rely heavily on the prediction in this setting, their behaviour is naturally similar.

For larger $\lambda$, as seen in Figures 4 and 8, the performance gap between the two algorithms increases. However, for $\lambda$ close to 1, both consistency and robustness approach $1/r\theta$. While the performance of $\mathsf{A}_r^1$ degrades significantly more slowly than that of $\mathsf{A}_r^0$ for $\lambda = 0.8$ (Figure 8), the values of $r$ and $1/r\theta$ remain close. Figure 4, presented in Section 5, is an intermediate

---

[4]If $[1, \theta]$ had been unbounded, $\mathscr{W}_p$ would only have been defined for distributions with a a $p^{\text{th}}$-moment.

setting between these two extremes, where $\mathsf{A}_r^1$ yields a better smoothness than $\mathsf{A}_r^0$, without having the values $r$ and $1/r$ close to each other.

