# OpenReview forum: "Pareto-Optimality, Smoothness, and Stochasticity in Learning-Augmented One-Max-Search"
_ICML.cc/2025/Conference — ICML 2025 poster_

### Official Review · Reviewer_oFaD · 2025-03-06

**Overall Recommendation:** 3

**Summary:**

This paper studies the learning-augmented one-max-search problem. In the classic one-max-search problem, the input of an algorithm is a sequence of prices $(p_i)_{i=1}^{n}\in [1,\theta]^n$. At each $i$, the algorithm must decide whether to accept a price $p_i$ and terminate, or to reject $p_i$ and proceed. In the learning augmented one-max-search problem, the algorithm has access to a prediction $y$ of the maximum price in $p$.

The objective of the algorithm is to achieve the best trade-off between consistency (the competitive ratio of the algo. if the given prediction $y$ is accurate), robustness (the competitive ratio of the algo. if the given prediction $y$ is arbitrarily inaccurate), and smoothness (the competitive ratio of the algo. as a function of the prediction error). This paper designs a pareto-optimal algorithm (w.r.t. the consistency and robustness tradeoff) that is also smooth, and prove that the smoothness of the algorithm matches the lower bound of any Pareto-optimal algorithm.

**Claims And Evidence:**

The claims are supported by the theoretical analysis.

**Essential References Not Discussed:**

This paper discussed essential related works.

**Experimental Designs Or Analyses:**

I checked the experiments in Section 5.

**Methods And Evaluation Criteria:**

Yes.

**Other Comments Or Suggestions:**

I have no other comments.

**Other Strengths And Weaknesses:**

Strengths:

-	The theoretical results in this paper are neat and solid.
-	The paper is well-written and self-contained.

Weakness:

-	The organization of this paper could be improved to include some high-level overview of proof techniques. While the difference between the algorithms proposed in this paper and prior algorithms is clear in Figure 1, the technical novelty in the analysis is not addressed.

**Questions For Authors:**

-	While Figure 2 clearly shows that the average performance when $\rho=1$ is better than $\rho=0$ (the algorithm discussed in prior work), the confidence fluctuation (shaded area) of the ratio also significantly overlaps. Is it because of the randomness of the environment? In addition, what is the relative performance of $\rho=1$ compared with $\rho=0$? E.g., for the same randomly generated environment, what is the win-rate of $\rho=1$ against $\rho=0$?
-	Can you also draw the theoretical bound given by Theorem 3.2 (or Theorem 4.6) in Figure 2? How well is the empirical performance of the algorithm characterized by the theoretical analysis?

**Relation To Broader Scientific Literature:**

The closest related works is Sun et al. (2021), which characterized the Pareto front of consistency and robustness. The idea of designing algorithms that have smooth performance w.r.t. the prediction error is an important question in learning-augmented algorithms.

**Theoretical Claims:**

I only checked the first 8 pages of this paper.

---

> ### Author Rebuttal · Authors · 2025-03-30
>
> We thank the reviewer for the time and effort spent on our submission and for the positive feedback. We address their questions and comments below.
>
> ### **Weakness**
> Due to page limitations, we could not always give detailed proof methodologies in the main paper. However, if the paper is accepted, we will use the additional page to enhance this aspect, as suggested by the reviewer. For instance, please see our response below for some specific paragraphs that we will add in regard to Section 3.
>
> **Technical Novelty.** Our techniques fundamentally differ from those used by Sun et al. While their work aims to determine the Pareto-optimal tradeoffs between consistency and robustness, our work extends their results by leveraging the knowledge of the Pareto-optimal frontier for consistency-robustness, and establishes smoothness guarantees, which is not addressed in the previous work. Specifically, we characterize all threshold algorithms that achieve this frontier and further analyze the optimal smoothness within this class of algorithms using two distinct notions of smoothness: multiplicative and additive.
>
> **High level overview of proof techniques.**
> - **Theorem 3.1**: If a threshold function $\Phi$ belongs to $\mathcal{P}_r$, then we use the bounds on $\Phi$ defining $\mathcal{P}_r$ to prove its Pareto-optimal robustness and consistency. Conversely, for a Pareto-optimal $\Phi$, we analyze the worst-case instances $I_n(q)$ (Equation 15), where prices increase uniformly from 1 to $q$ before dropping to 1. By evaluating $\text{ALG}/\text{OPT}$ on these instances for well-chosen $q$, we derive the bounds on $\Phi(y)$ that define $\mathcal{P}_r$.
> - **Theorem 3.2**: For any $\rho \in [0,1]$, the threshold of Algorithm $A^\rho_r$ lies in $\mathcal{P}_r$, ensuring $r$-robustness and $1/r\theta$-consistency (Theorem 3.1). To establish smoothness, we determine the best value of $s$ such that $\text{ALG}/\text{OPT} \geq (1/r\theta) \mathcal{E}^s$ across all price sequences and predictions. Since the threshold is piecewise-defined, it suffices to analyse cases based on $y$ and whether the maximum price is above or below the threshold. The final smoothness guarantee is then determined by the worst such case.
> - **Theorem 3.3**: We first prove that, on worst-case instances $I_n(q)$, any Pareto-optimal algorithm behaves like a threshold algorithm. Using results from Theorem 3.1, we establish that its threshold function must belong to $\mathcal{P}_r$. Finally, by applying the defining inequalities of $\mathcal{P}_r$ with well-chosen $q$ and $y$, we derive lower bounds on the smoothness $s$.
>
> ### **Questions**
> - **Confidence fluctuation.** The confidence fluctuation in Figure 2 is indeed a result of the randomness of the environment.
> - **Relative performance.** Could the reviewer clarify what they mean by the "win-rate" of $\rho=1$ against $\rho=0$? We interpret this as the ratio $CR(A^1)/CR(A^0)$, indicating how much $A^1$ gains compared to $A^0$. Both the empirical (solid line) and theoretical (dashed line) worst-case ratios are shown in the following figure (https://imgur.com/a/PmnrYOn). The figure shows that the empirical ratio is 1 when the prediction is perfect (i.e. $1-\mathcal{E} = 0$), as expected, since both algorithms have the same consistency. The ratio then increases abruptly to approximately $1.3$, for an arbitrarily small error, which is due to $A^0$'s brittleness. It then increases, up to a value of 1.5 approximately, after which it decreases down to 1, as the prediction becomes highly inaccurate.
>     This is explained, respectively, by the fact that $A^0$ remains inefficient for relatively small values of error, and by the fact that both algorithms have the same robustness.
> - **Theoretical bounds in the figure.** Thank you for the suggestion. We included the theoretical worst-case ratio (dashed lines) for $\rho \in$ {0, 0.5, 1} alongside the empirical ratios from Figure 2. The updated figure is available here (https://imgur.com/a/87xE5Qv). For the three values of $\rho$, the figure shows that the empirical ratio is better than the theoretical one because the latter represents the worst case over all predictions with error at most $\mathcal{E}$, while the empirical ratio is based on random predictions, explaining the observed gap. Note that, for $\rho < 1$, there is a sharp degradation in performance; for example, for $\rho=0$, this degradation occurs at infinitesimally small prediction error. In contrast, for $\rho=1$, the degradation is smooth, which aligns with our objectives and the intuition given in Section 3.

---

> > ### Comment · Reviewer_oFaD · 2025-04-01
> >
> > I appreciate the authors' response. I will maintain my positive evaluation and recommend incorporating the new figures and discussions in the revision.
> >
> > > Could the reviewer clarify what they mean by the "win-rate" of $\rho=1$ against $\rho=0$?
> >
> > I meant $\Pr(\mathbb{I}[CR(A^1) > CR(A^0)])$ where the probability is taken over the randomness of the environment? This is because confidence fluctuation in Figure 2 is significant due to the randomness of the environment, which makes the comparison a bit unclear. Another way to compare the algorithms is to look at their relative performance in the same environment, and then consider the randomness of the environment. The new figure already answered my questions.

---

### Official Review · Reviewer_qsuL · 2025-03-11

**Overall Recommendation:** 3

**Summary:**

This paper studies one-max search with prediction: Given a sequence of prices $p_1, \ldots, p_n$ in an online fashion and a prediction $y$ of the maximum price $p^*$, pick a price irrevocably to compete with $p^*$.  Previous algorithms with prediction can simultaneously achieve _consistency_ (when the prediction is correct, the performance is optimal) and _robustness_ (when the prediction is arbitrary, still achieves the worst-case-optimal competitive ratio).  There are also previous algorithms (Sun et al, 2021) achieving the optimal consistency-robustness tradeoff (_pareto-optimal_ in the two extreme cases).  However, those algorithms are not _smooth_: when the prediction accuracy degrades smoothly, the performance of the algorithm should change smoothly.  The authors provide the first Pareto-optimal and smooth algorithms for the problem of one-max search with predictions.  Moreover, this algorithm does not need randomization.

The technique is to fully characterize the class of _threshold algorithms_ that are Pareto-optimal, and then within this class find the best smooth algorithm.  The authors also show that the smoothness guarantee of the found algorithm is tight.

The authors then apply the main results to a setting where the prediction and the maximum price are sampled from some joint distribution, and study the impact of the distribution on the algorithmic performance.  They also provide experiments to show the advantage of their algorithm over a previous one.

**Claims And Evidence:**

The claims are supported by clear and convincing evidence, both theoretically and experimentally.

**Essential References Not Discussed:**

I don't know of essential missing references.

**Experimental Designs Or Analyses:**

I didn't find any issues in the experiments.

**(Strength 1)** On the positive side, the experiment with real-world Bitcoin data, showing that the proposed algorithm ($\rho = 1$) performs better than previous algorithm ($\rho = 0$), is interesting and strengthens this paper a lot in my opinion.

**Methods And Evaluation Criteria:**

The theoretical approach to this problem -- fully characterizing the class of Pareto-optimal algorithms and then selecting the best smooth algorithm among them -- is non-trivial and interesting.

**Other Comments Or Suggestions:**

(1) Page 6, second column of line 320: "this functional of $G$" -> "this functional of $F^*$" ?

(2) The last sentence before Section 4.1 "Finally, we show how to isolate the interaction of $G$ and $F^*$ using analytical tools from optimal transport theory" is confusing.  In Section 4.2, you isolate the effect of $G$ in Corollary 4.2 and isolate the effect of $F^*$ in Corollary 4.4, not using optimal transport theory.  Then in Section 4.3, you consider the joint effect of $G$ and $F$ -- you use the optimal transport theory here to find the worst coupling between $G$ and $F$ such that the competitive ratio attains the infimum.  I don't think you are using the optimal transport theory to "isolate the interaction of $G$ and $F$".

**Other Strengths And Weaknesses:**

I have mentioned three strengths above.

A **minor weakness**, in my opinion, is that this work is restricted to the one-max search problem.  The authors "believe their methodology can be applied to generalization such as the $k$-search problem" (Conclusion), but didn't give any theoretical evidence.

But since the one-max search problem is fundamental and this work gives an almost complete characterization for the Pareto-optimal and smooth algorithms for this problem, I am overall positive towards this work.

**Questions For Authors:**

(Q1) Does tight lower bound result (Theorem 3.3) hold for all randomized algorithms or just for deterministic algorithms?

**Relation To Broader Scientific Literature:**

**(Strength 2)** Previous algorithms for one-max search with predictions (Sun et al, 2021) achieve the optimal consistency-robustness tradeoff (_pareto-optimal_ in the two extreme cases).  However, those algorithms are not _smooth_.  Even though one-max search is such a fundamental question, it is still open whether a doubly optimal algorithm (Pareto-optimal and smooth) exists.  This work successfully answers this question by providing a doubly optimal algorithm.  Moreover, this algorithm is deterministic, which is even better. So, I think this is a significant contribution to the literature.

**(Strength 3)** The technical part of this work fully characterizes the class of Pareto-optimal algorithms for the one-max search problem with predictions.  This technical contribution might be of independent interest.

**Theoretical Claims:**

I checked the proof of Theorem 3.1 and briefly went over other proofs and didn't find any issues.

---

> ### Author Rebuttal · Authors · 2025-03-30
>
> We thank the reviewer for the time and effort spent on our submission and for their positive feedback. We address their questions below.
>
> **Minor weakness.**
> The $k$-search problem generalizes the one-max search problem, requiring a trader to sell $k$ items instead of one, with arbitrary prices in a bounded range $[p_{\min}, p_{\max}]$. Lee et al. studied this problem in a learning-augmented setting with a prediction of the maximum price, proposing a threshold-based policy that achieves a Pareto-optimal tradeoff between consistency and robustness, but lacks smoothness guarantees. Given the Pareto-optimal consistency and robustness values, we can aim to characterize all algorithms achieving them. Building on the intuition we established for one-max search, the smoothness of such algorithms depends on the maximum slope of their thresholds. We can then analyse the algorithm within this family with the smallest maximum slope, which we expect to have good smoothness guarantees.
>
> **Other Comments Or Suggestions**.
> (1) We thank the reviewer for pointing out the typo. We have corrected it.
> (2) We agree with the reviewer's observation and apologise for the confusing formulation. We meant to highlight the way that Optimal transport was used to characterise the effect of the interaction between $G$ and $F^*$ beyond their own effect. We will clarify this point.
>
> **Questions**.
> (Q1) The bound holds for deterministic algorithms. In the presence of randomization, the problem is equivalent to the one-way trading problem, in which fractions of an item can be bought. The interplay between smoothness and Pareto-optimality for one-way trading has been fully studied in "Overcoming Brittleness in Pareto-Optimal Learning-Augmented Algorithms" (Elenter et al., NeurIPS '24), where the authors proved that any algorithm achieving Pareto-optimal consistency and robustness is necessarily brittle (i.e., it cannot have any smoothness guarantees). This gives further evidence that, unlike many decision problems, the deterministic and randomized versions of one-max search are significantly different.

---

### Official Review · Reviewer_hcYC · 2025-03-11

**Overall Recommendation:** 4

**Summary:**

The authors study the one-max-search problem where the decision maker is presented with a stream of value within a range $[1, \theta]$. The decision maker has to irrevocably select and end the process, or forfeit a value in the stream. The authors study the consistency, and robustness tradeoff of a Pareto optimal algorithm that relies on a prediction of the optimal value.  Consistency refers to the approximation ratio of an algorithm when it is supplied with the correct prediction. Robustness refers to the approximation ratio of an algorithm supplied with incorrect prediction. The authors study the tradeoff as a function of the smoothness of the threshold function, where smoothness is defined using a symmetric and scale invariant multiplicative prediction error. They establish consistency of a specific class of threshold function that satisfies a given robustness $r$ and smoothness $\rho$. Furthermore, they show that their guarantees are nearly tight for any Pareto Optimal algorithm.  The authors also study the competitive analysis under multiple stochastic frameworks where price and/or prediction exhibit stochasticity.  Interestingly, when both exhibit stochasticity they show the authors establish performance guarantees using optimal transport.

**Claims And Evidence:**

The theoretical results stated in the paper is supported by proofs.

**Essential References Not Discussed:**

The important references seem present in my understanding.

**Experimental Designs Or Analyses:**

Numerical evaluations are presented. The results look good to me.

**Methods And Evaluation Criteria:**

The methods are standard for a theoretical paper.

**Other Comments Or Suggestions:**

Possible Improvements/Discussions:
- The current paper claims to establish a three-way Pareto optimality. However, in my understanding, for proper three-way relation the robustness should depend on the multiplicative error $\mathcal{E}$.
- On a related note to the above, the lower bound established relies on A being Pareto optimal for all $y$. What happens if we restrict the prediction to have bounded error  $\mathcal{E}_{\max}$.
- What is the relation of the optimal transport specified from a learning perspective? Can we discover the coupling in a offline and online learning manner?

**Other Strengths And Weaknesses:**

Strengths:
- This paper establishes a near optimal smoothness guarantees for Pareto optimal algorithms, and provides a simple threshold based algorithm that attains the nearly tight bounds.
- This paper studies the performance of the proposed algorithm, and establishes performance guarantees through optimal transport.

Weakness: Please check comments and suggestions.

**Questions For Authors:**

Please check comments and suggestions.

**Relation To Broader Scientific Literature:**

This paper establishes almost tight relationship between the consistency, robustness, and smoothness of an algorithm. This is a good addition to the understanding of the one-max-search problem.

**Theoretical Claims:**

I have checked the theoretical claims. I am reasonably certain about the correctness, but may have missed details (e.g. algebraic derivations of inequalities).

---

> ### Author Rebuttal · Authors · 2025-03-30
>
> We thank the reviewer for the time and effort spent on our submission, and for their positive feedback. We address their comments below.
>
> - The objective of the robustness term is to give a valid guarantee even under arbitrarily inaccurate predictions, since no assumption is made on the quality of the prediction. This is why the robustness term should not depend on the prediction error.
> -  It might be possible to improve the bound if the error is known to be at most $\mathcal{E}_{\max}$. However, our main objective is to design an algorithm that is able to leverage a prediction without any additional assumptions on its quality.  More generally, in any learning-augmented problem, if the error is assumed to be bounded, then the consistency, robustness, and smoothness guarantees might all be improved. While this question is out of the scope of our work, it constitutes a very interesting research direction.
> - Along a single trajectory of prices, it is impossible to learn the coupling as the whole trajectory provides only a single sample of the maximum price and a single prediction, which is statistically meaningless. However, it is possible to learn the coupling if the game is repeated several times using standard density estimation techniques, see e.g. (Silverman, 2018; Chen, 2017). Studying a repeated version of the one-max-search problem (whether on-- or offline) is a significantly methodologically different task. This raises an interesting further direction of research.
>
> **References.**
> Silverman, Bernard W. Density estimation for statistics and data analysis. Routledge, 2018.
> Chen, Yen-Chi. A tutorial on kernel density estimation and recent advances. Biostatistics \& Epidemiology 1.1 (2017): 161-187.

---

> > ### Comment · Reviewer_hcYC · 2025-04-03
> >
> > I thank the authors for the response. I agree that the posed questions are possibly out-of-scope. Maybe adding those to future works can improve the paper. I will maintain my positive score.

---

### Official Review · Reviewer_S3x7 · 2025-03-13

**Overall Recommendation:** 4

**Summary:**

This paper studies one-max-search problem in the learning-augmented setting and develops an algorithm that is both Pareto-optimal and smooth to the multiplicative prediction error.

**Claims And Evidence:**

The claims made in the submission are supported by clear and convincing evidence.

**Essential References Not Discussed:**

NA

**Experimental Designs Or Analyses:**

I checked the soundness/validity of experimental designs or analyses.

**Methods And Evaluation Criteria:**

The proposed methods and/or evaluation criteria (e.g., benchmark datasets) make sense for the problem or application at hand. The experiments clearly demonstrate the improvements of the proposal algorithm compared to existing methods.

**Other Comments Or Suggestions:**

NA

**Other Strengths And Weaknesses:**

The paper characterizes all Pareto-optimal thresholds in a general class which captures many previous works. This result seems to be fundamental to the one-max-search problem.

**Questions For Authors:**

In the stochastic setting, can we have high-probability guarantees on ALG?

**Relation To Broader Scientific Literature:**

The key contributions of the paper is related to general ML.

**Theoretical Claims:**

I checked the correctness of the arguments appeared in the main paper.

---

> ### Author Rebuttal · Authors · 2025-03-30
>
> We thank the reviewer for the time and effort spent on our submission, and for their positive feedback.
>
> In proposing a stochastic formulation of the one-max-search problem with predictions, we aimed to introduce the concept of a stochastic notion of accuracy for predictions. We gave bounds in expectation, but, as the reviewer suggests, one can also consider high-probability bounds. However, these bounds would concern the joint tail probabilities of $P^*$ and $Y$, which are difficult objects to conceptualise. In contrast, bounds in expectation naturally give rise to a characterisation of the worst case as an optimal transport problem.

---

### Decision · Program_Chairs · 2025-05-01

**Decision:**

Accept (poster)

**Comment:**

The paper studies the one-max search problem in the presense of machine-learned predictions or advice. Without advice, the stopping problem of picking the (approximately) maximum value in a sequence of revealed values, when the values are known to be in the interval $[1,\theta]$ is known to admit a deterministic algorithm with a tight $1/\sqrt{\theta}$ approximation. In the presence of advice, prior work established the Pareto frontier between two important quantities of interest in such problems, namely, consistency and robustness, and the value of $\theta$ for this problem. What this paper is interested in is, to not only have the best tradeoff between consistency and robustness, but also have a "smooth" algorithm, namely, one whose performance decays smoothly as the prediction error grows. Prior to this work, the existence of an algorithm that is simultaneously Pareto-optimal and smooth was open, and this work answers that question positively.

The committee appreciated (a) the tight results in this work, (b) the applications to the stochastic setting where the predictions and prices both stem from a random environment and this paper shows how the performance depends on the distributions of predictions and prices, and their coupling and (c) the technical content of this work, bringing in techniques from optimal transport.

Overall, this is a solid contribution to ICML.